# Scar matrix drives Piezo1 mediated stromal inflammation leading to placenta accreta spectrum

Du Wenqiang [1,2], Ashkan Novin [1,2], Yamin Liu[2], Junaid Afzal [3], Yasir Suhail[1,2], Shaofei Liu[1,2], Nicole R. Gavin[4], Jennifer R. Jorgensen[4], Christopher M. Morosky[4], Reinaldo Figueroa[5], Tannin A. Schmidt[1], Melinda Sanders[4,6], Molly A. Brewer[4] & Kshitiz [1,2] ✉

Scar tissue formation is a hallmark of wound repair in adults and can chronically affect tissue architecture and function. To understand the general phenomena, we sought to explore scar-driven imbalance in tissue homeostasis caused by a common, and standardized surgical procedure, the uterine scar due to cesarean surgery. Deep uterine scar is associated with a rapidly increasing condition in pregnant women, placenta accreta spectrum (PAS), characterized by aggressive trophoblast invasion into the uterus, frequently necessitating hysterectomy at parturition. We created a model of uterine scar, recapitulating PAS-like invasive phenotype, showing that scar matrix activates mechanosensitive ion channel, Piezo1, through glycolysis-fueled cellular contraction. Piezo1 activation increases intracellular calcium activity and Protein kinase C activation, leading to NF-κB nuclear translocation, and MafG stabilization. This inflammatory transformation of decidua leads to production of IL-8 and G-CSF, chemotactically recruiting invading trophoblasts towards scar, initiating PAS. Our study demonstrates aberrant mechanics of scar disturbs stroma-epithelia homeostasis in placentation, with implications in cancer dissemination.

Scar tissue formation as a result of surgery, injury, trauma, or infarction is a hallmark of adult wound repair, resulting in profound changes in the tissue microenvironment[1,2]. Pathological scarring by surgery or trauma occurs through progressive remodeling of the granulation tissue, and is characterized by high type I collagen content, decreased cellularity, high mechanical rigidity, and aberrant matrix ultrastructure[3,4]. Although mechanisms of scar formation have been extensively studied[5–8], relatively less is understood about the effect of existing scar or fibrosis on tissue homeostasis.

To understand the mechanisms driving scar matrix-induced dysregulation in tissue homeostasis, we sought to investigate a disease attributable directly to a standard and commonly occurring preexistent scar: placenta accreta spectrum (PAS). PAS is a general definition for pathologies characterized by abnormally invasive placentation (accreta, increta, and percreta)[9–11], and is primarily considered to be an outcome of pre-existent uterine scar from cesarean surgery[12–15] (Fig. 1A). Uterine scars due to cesarean sections have increased rapidly, accounting for 21% of all childbirth worldwide[16,17]. PAS has also

[1]Department of Biomedical Engineering, University of Connecticut Health Center, Farmington, CT, USA. [2]Department of Biomedical Engineering, University of Connecticut, Storrs, CT, USA. [3]Division of Cardiology, Department of Medicine, University of California San Francisco, San Francisco, CA, USA. [4]Department of Obstetrics and Gynecology, University of Connecticut Health Center, Farmington, CT, USA. [5]Department of Obstetrics and Gynecology, Saint Francis Hospital and Medical Center, Hartford, CT, USA. [6]Department of Pathology, University of Connecticut Health Center, Farmington, CT, USA. ✉ e-mail: kshitiz@uchc.edu

**Fig. 1 | In vitro model of PAS recapitulates the abnormally deep trophoblastic invasion. A** Schematic showing abnormally deep invasion of extravillous trophoblasts (EVTs) in the decidual stroma proximal to scar. SCT: syncytiotrophoblast, CT: cytotrophoblast, dF: decidual fibroblast, Scar-dF: transformed dFs proximal to scar, Physio-dF: normal dFs distal to scar, V: placental villi, M: myometrium. **B** H&E and immunohistochemistry images of maternal-fetal interface tissue sections from PAS patients showing HLA-G⁺ EVT (green), and Vimentin labeled dFs (red); nuclei marked with DAPI (blue). $n = 6$ biological replicates. **C** Picrosirius red staining of tissue sections from regions proximal, and distal to pre-existent scar, with orientation distribution of collagen fibers from different PAS patients quantified in (**D**). $n = 3$ biological replicates. **E** Surface topography of Scar matrix imaged by atomic force microscopy (AFM) in PBS; **F** Photo and spatial rigidity characterization of normal endometrial tissue; Graph (bottom) shows mean rigidity of endometrial tissue, Physio, and Scar matrices. $n = 8$, 25, and 8 biologically independent experiments. $p = 3 \times 10^{-42}$. **G** Schematic showing workflow to establish in vitro Scar induced PAS model with distinct invasion assays. **H, I** Phase contrast image showing in-situ HTR8 spheroid invasion into ESFs decidualized on Physio; Graphs showing

line integral convolution representation of HTR8 invasion into ESFs decidualized on Physio, or Scar matrices. **I** Normalized HTR8 invasion area ($S/S_0$) as a function of invasion time. $n = 10$ biological replicates. **J** Fluorescent images of HTR8 (red) invasion into ESFs pre-decidualized on Physio, or Scar at time 0, and 24 h; Graph showing aerial invasion normalized to initial interface length; $n = 8$ interfaces; $p = 0.005$. **K** Apotome scanning of HTR8 spatial nuclear locations relative to dESF monolayers 72 h after invasion; **L** Quantification of individual HTR8 distance to dESF monolayer; $n = 255$ cells; $p = 6 \times 10^{-79}$. **M** Volcano plot showing differentially expressed genes in dESFs on Scar and Physio. **N** Ingenuity Pathway Analysis based prediction of activated transcription factors in dESFs on Scar and Physio; $n = 3$ biological replicates. Data in all bar graphs are showing as mean ± s.d.; statistical significance is determined by unpaired two-tailed t-test (**$p < 0.01$, ****$p < 0.0001$, and ns not significant). Source data are provided as a Source Data file. **A, G** created with BioRender.com released under a Creative Commons Attribution-NonCommercial-NoDerivs 4.0 International license (https://creativecommons.org/licenses/by-nc-nd/4.0/deed.en).

increased in parallel, rising from about 1 in 30,000 in 1950s to about 1 in 300 pregnancies in the US alone[18–20]. Therefore, uterine scar-induced PAS could be considered as a model to investigate scar-induced pathology. PAS results in serious maternal and fetal complications, including life-threatening hemorrhage, necessitating hysterectomy. PAS may also result in damage to adjacent organs, maternal mortality, and preterm birth, resulting in exorbitant clinical, psychological, and economic costs[21,22]. Although the most important predictor for PAS is previous uterine scar due to cesarean delivery, surgeries or curettage, or other damage to the uterine wall, little is understood about the mechanisms driving scar-induced PAS onset and progression.

Placentation in humans involve deep invasion of fetal extravillous trophoblasts (EVTs) into the maternal endometrial stroma[23]. In anticipation of placentation, the maternal endometrial stroma undergoes a profound transformation, termed decidualization. We, and others have shown that decidualization is, in part, a mechanism to acquire resistance against EVT invasion[24,25]. The decidua has to both support the deep maternal invasion by EVTs, as well as prevent excessive invasion potentially leading to PAS-like pathology. Physiological placentation is a tightly regulated phenomenon which requires careful balancing between the pro- and anti-invasive mechanisms, resulting in a negotiated homeostasis for optimal invasion[26,27]. This negotiation is partly arrived at by molecular communication between fetal and maternal cells[28].

Although very little is known about the mechanisms driving the progression of PAS, the current prevailing hypothesis explaining the PAS pathogenesis is that the trophoblasts may preferentially invade into the acellular scar tissue left by a previous cesarean surgery or other trauma to the uterine wall. We posit counter arguments to this thesis. If cell permeable collagen tracks were present in acellular collagenous scar, other cells within the decidua could enter and re-cellularize the region. Instead, the absence of cells in the highly collagenous acellular region of scar proper indicates a physical barrier to cell invasion. Based on these considerations, we posit that the scar may transform the decidua in its proximity, promoting aggressive trophoblast invasion. Despite the frequent absence of decidua at the invasion sites in the third trimester of PAS cases, histological examination indicates endometrial stroma overlies the cavity of scar after cesarean sectioning[29–34]. We hypothesize that uterine scarring leads to a failure of normal decidualization in its proximity, which causes abnormal interaction between the decidua and the EVTs. This aberrant communication between decidua and EVTs leads to failure of maternal tissues to restrain the invading trophoblasts.

Based on histological characterization of maternal-fetal interface from PAS patients, we created a model of scar-decidua encapsulating the essential microenvironmental features of a uterine scar. We found that endometrial stromal fibroblasts (ESFs) obtained from normal patients exhibited dysregulated decidualization on Scar matrix, and recapitulated the phenotype of deep trophoblast invasion. We present here the mechanisms by which mechanical signals from the scar matrix modulate decidual fibroblasts to produce inflammatory cytokines recruiting EVTs towards the scar. Specifically, we show that this inflammatory transformation of dESF is achieved by Scar matrix-induced activation of mechanosensitive ion channel, Piezo1, resulting in increased intracellular calcium activity that leads to Protein kinase C (PKC) activation. PKC activation results in NF-κB phosphorylation, stabilization of a small Maf protein, MafG, and onset of inflammation-related transcription. In addition, we also found that scar transformed dESFs to a high contractile force-producing state, fueled by glycolysis, further enhancing PAS-like manifestation of deep placental invasion. The phenotypes were similar to the promotion of cancer dissemination by cancer-associated fibroblasts (CAFs)[35–37], suggesting that the mechanisms we identify are implicated in other scar or fibrosis-related pathologies, including cancer metastasis, and wound healing.

## Results

### Creating decidua-trophoblast interface on substrate mimicking uterine scar

Histological analysis of PAS tissues from multiple patients showed deep invasion of placental villi into the myometrium, marked by disseminated HLA-G[+] extravillous trophoblasts (EVTs) (Fig. 1A, B). Collagenous regions, which were largely acellular, showed highly aligned collagen fibers in the region proximal to scar (Fig. 1C, D and Supplementary Fig. 1A). Anisotropy of collagen fibers has been observed in scars of other tissues, including the ovary and endometrium by second harmonics imaging[38].

We asked if the aberrant mechanical signals present at the scar transform decidual fibroblasts to be more invasable to EVT invasion. To answer this question, we sought to create a scar-like model of decidua, and ask if it manifests PAS-like phenotype characterized by enhanced EVT invasion. To mimic the directional collagen bundles present in scar, we patterned an anisotropic nanowrinkled pattern on the hydrogel, with individual pitch matching those of collagen fibrils in fibrotic tissue, confirmed by atomic force microscopy (Fig. 1E and SupplementarySupplementary Fig. 1B). For scar-mimetic matrix (Scar), we created polyacrylamide hydrogel substrate rigidity matched to those reported in elastography data[39] (Fig. 1F). For the control physiological matrix (Physio), we mechanically profiled normal, healthy endometrium, and created a flat substrate mimicking the tissue rigidity (Fig. 1F and Supplementary Fig. 1C, D). Physio was conjugated with 40 µg/ml type I collagen, while anisotropic high rigidity Scar was conjugated with 100 µg/ml type I collagen. ESFs isolated from normal patient endometrial biopsies were seeded on Physio and Scar, and decidualized for 4 days before downstream experimentation (Fig. 1G).

### Scar matrix transforms decidual fibroblasts to chemotactically recruit extravillous trophoblasts

We have previously shown that stromal permissibility to invasion, both in decidua and in cancer stroma, is an evolved phenotype in mammals, with stromal fibroblasts being active players in determining their own invasibility[40]. Indeed, while decidualization is an evolved response to resist trophoblast invasion, EVTs cooperate with decidual fibroblasts via intercellular paracrine signaling to assist in their own invasion[41]. We asked if EVTs invade more on Physio or Scar decidua using multiple invasion assays designed to elicit different plausible mechanisms regulating stromal invasion by EVTs (Fig. 1G).

First, we observed spheroids created from HTR8, a cell line derived from EVTs, measuring their spreading and dissemination on a layer of ESFs decidualized either on Physio or Scar substrate using live cell phase contrast microscopy (Fig. 1H and Supplementary Movie 1). Particle Image Velocimetry (PIV) analysis of this early placentation model revealed that Scar decidua significantly promoted HTR8 spreading and invasion (Fig. 1H, I, SupplementarySupplementary Fig. 1E, and Supplementary Movie 2). We then asked if the Scar matrix itself was causal in increased HTR8 invasion, or if it transforms dESFs to be more invasable to HTR8 invasion. We therefore decidualized ESFs on Physio or Scar, suspended them after trypsinization and patterned them in a monolayer juxtaposed to a monolayer of H2B-mCherry labeled HTR8, the interface being orthogonal to an underlying nanopatterned substrate. We have used this platform, termed Accelerated Nanopatterned Stromal Invasion Assay (ANSIA) to measure stromal invasibility with high sensitivity[24,42]. ANSIA eliminates the variation in invasion by preexisting cell orientation in the monolayer, as well as accelerates the invasion phenotype by unidirectional alignment of cellular actomyosin assemblies, facilitating rapid quantitative screening of stromal parameters. Time course analysis on ANSIA showed that dESFs decidualized on scar exhibited reduced resistance to HTR8 invasion (Fig. 1J, Supplementary Fig. 1F-G, and Supplementary Movie 3).

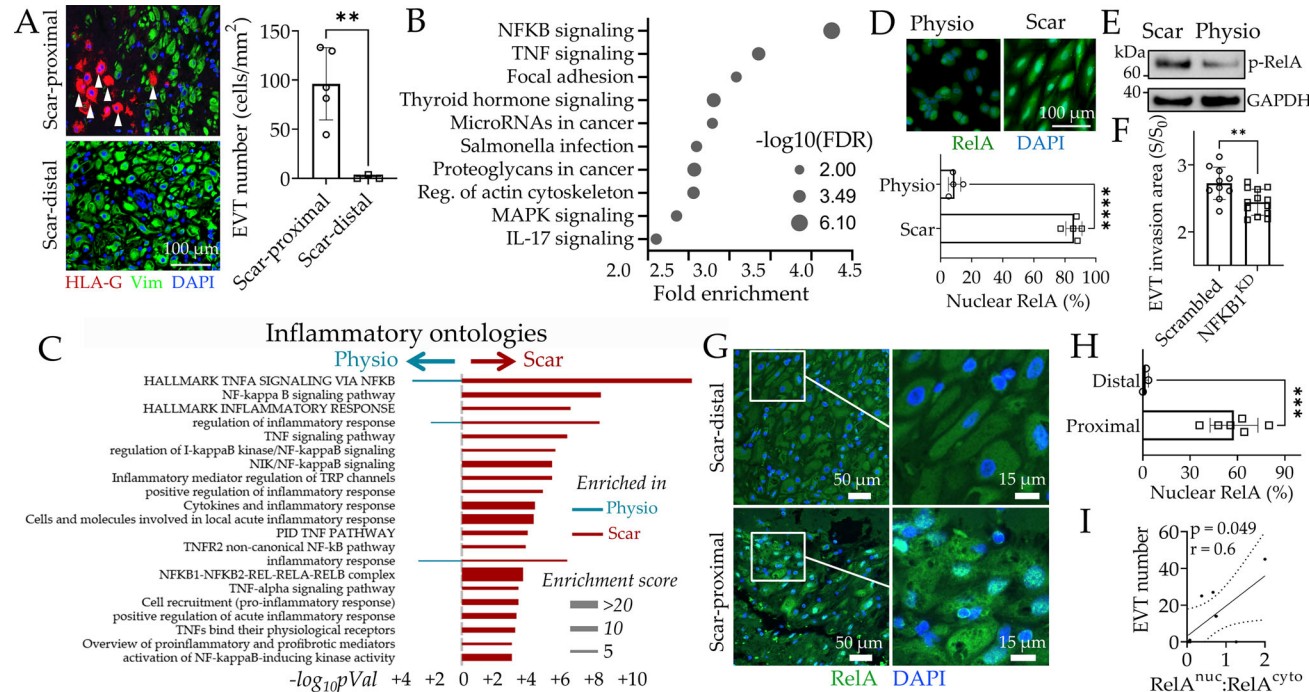

**Fig. 2 | Scar transforms decidual fibroblasts into NF-κB mediated inflammatory state. A** Representative immunohistochemistry images of maternal-fetal interface (MFI) tissue sections from PAS patients showing EVTs present in decidual regions proximal, and distal to collagenous acellular scar regions; Quantification of EVT density in either region in right panel; $n = 5$ and 3 for regions proximal and distal to scar respectively; EVTs and dESFs are marked with HLA-G (red; arrow heads) and Vimentin (green), respectively. $p = 0.005$. **B** KEGG pathway enrichment analysis showing signaling pathways differentially enriched in dESFs on Scar and Physio; **C** Gene ontologies related to inflammation enriched in dESFs on Scar and Physio. **D** Representative immunofluorescence images showing RelA (p65) location in dESFs on Physio and Scar; Quantification showing percentage of dESF with nuclear RelA in lower panel. $n = 4$ and 5 fields of view for Physio and Scar, respectively. $p = 5 \times 10^{-8}$. **E** Immunoblot showing abundance of phosphorylated RelA (p-RelA) in dESFs on Physio and Scar. Experiments are repeated twice with

similar results. **F** ANSIA based analysis of stromal invasion of primary EVTs into dESF compartment with scrambled, or gene silenced for NFKB1; $n = 10$ and 13 locations for scrambled and NFKB1$^{KD}$, respectively. $p = 0.005$. **G** Representative immunohistochemistry images of MFI tissue sections from PAS patient showing RelA intracellular localization in decidual regions proximal, and distal to scar; **H** Quantification of percentage of decidual fibroblasts with nuclear RelA; $n = 3$ and 6 locations for distal and proximal, respectively. $p = 0.0005$. **I** Pearson correlation test shows Pearson coefficient ($r$) of EVT number per field of view, and ratio of decidual fibroblasts with nuclear and cytoplasmic RelA and total decidual fibroblasts in PAS MFI tissue sections; a two-tailed $p$-value for Pearson's $r$ is calculated; $n = 8$ field of views. Data in figures **A**, **D**, **F**, **H** are showing as mean ± s.d.; statistical significance is determined by unpaired two-tailed $t$-test (**$p < 0.01$, ***$p < 0.001$). Source data are provided as a Source Data file.

Stromal invasion is a complex process and is a composite outcome of several potential sub-phenotypes, including migration of epithelial cells[43], stromal matrix degradation[44], mechanical coupling between epithelial and stromal cells[37], paracrine recruitment of epithelial cells, breach of cell-cell adhesion[45], etc. As decidualization on Scar resulted in more invasable dESFs, we asked if paracrine signals from these Scar-altered dESFs contribute to increased HTR8 invasion. We first spaced trophoblast spheroids away from the dESFs monolayer with a layer of collagen gel before the initiation of invasion (Fig. 1G and Supplementary Fig. 1H), and then recorded the relative nuclear spatial position of trophoblasts to dESFs monolayer after two days of invasion using structured illumination (Apotome) based imaging. Our results showed that trophoblasts are closer to dESFs on Scar than dESFs on Physio matrices (Fig. 1K, L). These data suggested that ESFs decidualized on Scar produce cytokines that recruit HTR8s towards scar (Fig. 1L). To gain insight into the underlying mechanisms contributing to the aberrant EVT invasion, we isolated the RNA from dESFs on Physio and Scar for sequencing. We found large differences in gene expression (Supplementary Fig. 1I-J and Supplementary Fig. 2), showing a marked reduction in decidual marker genes like *IGFBP1*, as well as increase in inflammatory cytokine CXCL8 (Fig. 1M). TransFac predicted activation of transcription factors (TFs) highlighted inflammatory TFs, including those belonging to the NF-κB pathway (*NFKB1*, *RELA*, *NFKBIA*), *HDAC1*, which converts fibroblast into cancer-associated fibroblasts (CAFs)[46], *RUNX1* which prognosticates immune infiltrate in

CAFs[47], as well as *NEF2L2* which encodes the key antioxidant TF Nrf2, also known to promote metastasis in CAFs[48] (Fig. 1N). Overall, our data suggest that mechanical cues presented by the scar matrix alter dESF state to chemotactically recruit EVTs preferentially towards the scar.

## Scar promotes NF-κB-driven dESFs inflammation

Immunocytochemistry staining of tissue slides from PAS patients revealed that many HLA-G$^+$ EVTs were present in the decidual region proximal to the collagenous scar vs decidual location distal from the scar, consistent with our PAS model (Fig. 2A and Supplementary Fig. 3A). KEGG pathway analysis on differentially expressed genes in Scar decidua confirmed increased activation of inflammatory pathways including NF-κB, a master regulator of immune activation, and IL-17, as well as mechanotransduction pathways associated with focal adhesion and actin cytoskeletal regulation (Fig. 2B and Supplementary Fig. 3B). A broader gene set analysis revealed activation of several inflammation related ontologies on Scar, the top being those related to tumor necrosis factor alpha (TNFA) and NF-κB signaling (Fig. 2C). Gene set enrichment analysis, a non-parametric statistical test, also showed high enrichment of inflammatory response on Scar vs Physio, as well as TNFα signaling via NFKB (Supplementary Fig. 3C). Activation of NF-κB associates with the accumulation of RelA subunit in the nucleus, driving transcription of several inflammatory genes. Immunofluorescence confirmed increased RelA localization in the nuclei on ESFs decidualized on Scar vs Physio (Fig. 2D and Supplementary

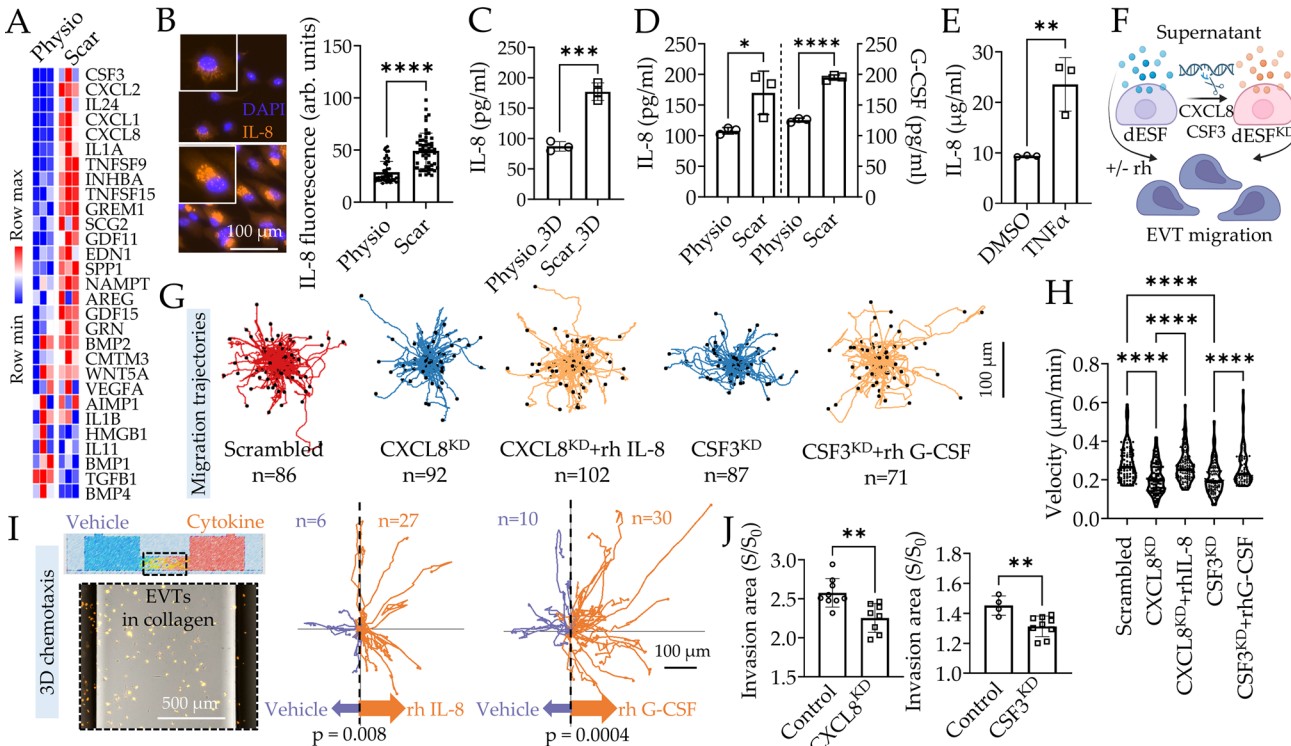

**Fig. 3 | IL-8/G-CSF secreted by Scar transformed decidual fibroblasts chemotactically recruit EVTs. A** Heatmap showing significant differential ligand encoding genes expressed in dESFs on Scar and Physio matrices. **B** Representative IL-8 immunofluorescence images of dESFs treated with protein transport inhibitor GolgiStop for 6 h on Physio and Scar matrices, quantification shown in right panel; $n = 49$ and 53 cells for Physio and Scar, respectively. $p = 5 \times 10^{-11}$. **C, D** ELISA based analysis of IL-8 and G-CSF concentration in supernatant of dESFs on Physio and Scar in 3D and 2D; n = 3 samples. $p = 0.0007$ in (**C**) and p = 0.04 and $2 \times 10^{-5}$ in (**D**). **E** ELISA based measurement of IL-8 concentration in supernatant of dESFs treated overnight with DMSO, or 100 ng/ml TNFα; $n = 3$ samples. $p = 0.0097$. **F** Experimental workflow to test migration of HTR8 in medium conditioned from dESFs with gene silenced for IL-8 and G-CSF encoding genes, CXCL8 and CSF3, respectively. **G** Migration trajectories (initial location (x, y = 0,0)) of HTR8 conditioned with medium from dESFs silenced for CXCL8 and CSF3 genes, without or with addition of recombinant human (rh) IL-8 and G-CSF; Quantification of averaged velocities over 24 h shown in (**H**); p = $1 \times 10^{-8}$, $8 \times 10^{-7}$, $8 \times 10^{-9}$, and $6 \times 10^{-5}$; n listed below each condition. **I** 3D chemotaxis of primary EVTs in collagen gel towards IL-8 and G-CSF gradient; Shown is a representative image of EVTs in collagen gel (left); Trajectories of individually tracked EVTs from their initial location (0,0) (middle and right); Cell trajectory with mean displacement towards cytokine end are labeled red, and counted (n); p value showing Rayleigh test of cell trajectories: $p < 0.05$ is considered chemotaxis. **J** ANSIA-based stromal invasion analysis of HTR8 in monolayer of dESFs silenced for CXCL8 and CSF3 genes; Control refers to scrambled sgRNA. $p = 0.003$ and 0.005. Data in figures B−E, H and J are showing as mean ± s.d.; statistical significance is determined by unpaired two-tailed t-test (*$p < 0.05$, **$p < 0.01$, ***$p < 0.001$, and ****$p < 0.0001$). Source data are provided as a Source Data file. Figure 3F created with BioRender.com released under a Creative Commons Attribution-NonCommercial-NoDerivs 4.0 International license (https://creativecommons.org/licenses/by-nc-nd/4.0/deed.en).

Fig. 3D), also confirmed for phosphorylated RelA abundance (Fig. 2E). We then sorted HLA-G⁺ cells from decidua basalis (Supplementary Fig. 3E) and functionally tested the effect of NF-κB activation in regulating invasibility of dESFs using ANSIA. We have previously shown that *NFKB1* increases invasibility of skin fibroblasts to cancer invasion[49]. *NFKB1* silencing significantly decreased dESF resistance on Scar-like matrix, highlighting its role in regulating stromal response to epithelial invasion (Fig. 2F and Supplementary Movie 4). Histology of PAS tissue with residual decidua showed that decidual fibroblasts in region proximal to acellular collagenous scar-like location mostly exhibited nuclear localization of RelA, key indicator of NF-κB activation. (Fig. 2G and Supplementary Fig. 3F−H). In contrast, distal decidua had very few decidual fibroblasts with nuclear RelA localization (Fig. 2G, H). We also found the nuclear colocalization of p50 with RelA in scar-proximal decidual fibroblasts (Supplementary Fig. 3G). Furthermore, we found a strong correlation between EVT density in decidual regions with nuclear localization of RelA in decidual fibroblasts (Fig. 2I), highlighting that NF-κB activation in decidual fibroblasts promoted EVT recruitment. Additionally, our pathway enrichment analysis of a recently available PAS single-cell RNA sequencing (scRNAseq) data[50] showed that NF-κB pathway is indeed highly enriched in adherent decidua of PAS patients (Supplementary

Fig. 3I, J). Overall, these data showed that scar matrix transformed the decidualization of ESFs in its proximity to a more inflammatory state, mediated by NF-κB, promoting more than optimal EVT invasion.

## dESFs on Scar matrix chemotactically recruit EVTs via secreted IL-8 and G-CSF

To identify potential paracrine signals, which induced dESFs on Scar to recruit EVTs, we searched for ligand encoding genes differentially regulated between Physio and Scar dESFs, finding several inflammatory cytokines in the list (Fig. 3A). These genes included *CSF3* which encodes granulocyte colony-stimulating factor G-CSF, several immunocyte recruiting C-X-C motif family ligands including *CXCL1*, *CXCL2*, and *CXCL8*, the latter encoding for IL-8, other interleukins IL-1A, IL-1B and IL-11, and several ligands of the tumor necrosis factor superfamily (Fig. 3A). It has been reported that IL-8 and G-CSF can regulate trophoblast migration in various contexts[51,52], and so we tested if dESFs on Scar may recruit EVTs via IL-8 and G-CSF. We confirmed increased abundance of IL-8 in dESFs both in 2D Scar model using immunostaining (Fig. 3B and Supplementary Fig. 4A), and in 3D Scar model with ELISA (Fig. 3C and Supplementary Fig. 1H), compared to respective Physio models. We also confirmed increased IL-8 and G-CSF secretion on 2D Scar using ELISA (Fig. 3D). We then asked if NF-κB regulated IL-8

and G-CSF production. TNFα treatment of dESFs significantly increased IL-8 secretion (Fig. 3E). To test if dESFs produced IL-8 and G-CSF could influence EVT migration, we tracked H2B-mcherry labeled HTR8s using live epifluorescence microscopy in the presence of conditioned medium from dESFs on Scar, silenced for genes encoding IL-8, G-CSF, or scrambled control. We verified that gene silencing of *CXCL8* showed no effect on the inflammatory phenotype of dESFs by assessing the expression of α-SMA and vimentin (Supplementary Fig. 4B–E). Conditioned medium from dESF^CXCL8-KD significantly reduced HTR8 displacement, which was reversed on addition of recombinant human IL-8 (Fig. 3F–H, Supplementary Fig. 4F, and Supplementary Movie 5). A similar effect was observed for conditioned medium from dESFs silenced for *CSF3* (dESF^CSF3-KD), which significantly reduced HTR8 velocity, while addition of recombinant G-CSF increased it again (Fig. 3F–H and Supplementary Fig. 4F).

The presence of a scar is a spatially localized presentation of mechano-chemical stimuli. We therefore asked if cytokines secreted by dESFs at the scar can chemoattract EVTs through the endometrial stroma towards its source, the Scar decidua. Using a microfluidically generated gradient of IL-8 and G-CSF on primary EVTs and HTR8 embedded in a 3D collagen gel, we tracked their displacement over time. We found that both exhibited a strong displacement bias parallel to IL-8, as well as G-CSF gradients (Fig. 3I, Supplementary Fig. 4G, and Supplementary Movie 6). Finally, we used ANSIA to quantitatively measure collective EVT invasion into dESFs. Our ANSIA results showed that gene silencing of *CXCL8* and *CSF3* significantly reduced dESF invasibility (Fig. 3J and Supplementary Fig. 4H).

## Piezo1 activation on Scar decidua increases NF-κB mediated secretion of IL-8 and G-CSF

A scar results in a significant change in the mechanical milieu of decidual stroma, to which decidual fibroblasts are likely to respond differently than to the normal matrix. Indeed, immunocytochemistry analysis of PAS patients' tissue sections showed that dESFs with nuclear RelA were located closer to the acellular areas compared to dESFs with cytoplasmic RelA, indicating that scar matrix likely drove nuclear translocation of RelA in decidual fibroblasts (Fig. 4A and Supplementary Fig. 5A). Because cells sense extracellular mechanical cues through a class of proteins on the plasma membrane known as mechanosensitive ion channels (MSICs)[53], we first asked if dESFs express any MSICs. Gene expression analysis revealed that *PIEZO1*, which encodes a key mechanosensitive ion channel, Piezo1, was the only MSIC highly expressed and significantly upregulated on Scar vs Physio dESFs (Fig. 4B). Our analysis of PAS scRNAseq data showed that *PIEZO1* expression is indeed upregulated in adherent decidua of PAS patients (Supplementary Fig. 5B). ANSIA based quantitative analysis of dESF invasibility showed that *PIEZO1* knockdown significantly increased dESF resistance to primary EVT spheroids and HTR8 invasion (Fig. 4C, Supplementary Fig. 5C-F, and Supplementary Movie 7). To further confirm the role of *PIEZO1* in regulating dESF invasibility, we performed EVT spheroids invasion in Matrigel plugs embedded with wildtype and gene edited dESFs in mouse (Fig. 4D, Supplementary Fig. 6). Similar to our in vitro invasion assay, we found that *PIEZO1* knockdown significantly increased dESFs resistance to HTR8 spheroid invasion, suggesting Piezo1's causality in regulating PAS-like phenotype of increased decidual invasibility (Fig. 4E, Supplementary Fig. 6). Piezo1 is a membrane stretch gated $Ca^{2+}$ permeable channel[54]; therefore, we tested if Scar matrix results in altered calcium dynamics within dESFs using genetically encoded calcium indicator unless otherwise mentioned. Scar matrix induced significantly more frequent, and higher amplitude $Ca^{2+}$ oscillations (Fig. 4F–H, Supplementary Fig. 5G-H, and Supplementary Movie 8). To confirm that these increased $Ca^{2+}$ levels and oscillations on Scar were Piezo1 mediated, we measured $Ca^{2+}$ dynamics upon treatment of Yoda1, a potent Piezo1 activator. Yoda1 resulted in significantly higher basal levels of $Ca^{2+}$ intensity (Fig. 4I and

Supplementary Movie 9). In contrast, gene silencing for *PIEZO1* in ESFs, and then decidualization on Scar resulted in nearly complete cessation of calcium oscillations (Fig. 4J). Piezo1 is classically described to be activated by changes in membrane tension[55]. We therefore asked if directly perturbing cellular membrane properties could regulate calcium signaling in dESFs. When dESFs were treated with methyl-β-cyclodextrin (MβCD), which chelates cholesterol from lipid rafts and increases membrane stiffness[56], we observed a dramatic increase in frequency of $Ca^{2+}$ oscillations (Supplementary Fig. 5I).We then asked if Piezo1 activation in dESFs on Scar could regulate NF-κB activation by checking RelA phosphorylation, a key regulator in NF-κB activation by enhancing its transactivation potential[57,58]. Immunoblot showed that phosphorylated RelA was reduced in dESFs treated with Piezo1 inhibitor, GsMTx-4, while Piezo1 activator Yoda1 increased it (Fig. 4K). Since GsMTx-4 also inhibits calcium activities gated by other TRP channels, we further confirmed the causal link between Piezo1 and NF-κB activation using CRISPR/Cas9 gene silencing in ESFs followed by decidualization. As expected, Piezo1 gene silencing resulted in decreased phosphorylated RelA abundance (Fig. 4K). Furthermore, ELISA showed that Yoda1 increased IL-8 and G-CSF production, while GsMTx4 significantly decreased production of both cytokines (Fig. 4L). Gene silencing of *PIEZO1* also markedly reduced IL-8 and G-CSF production in dESFs on Scar (Fig. 4M). Remarkably, ELISA also showed that direct perturbation of membrane stiffness by MβCD resulted in significant increase in IL-8 production, as well as of G-CSF (Supplementary Fig. 5J). These data show that increased membrane tension of dESFs on Scar matrix can transform the fibroblasts into an inflammatory state, producing chemotactic cytokines for EVTs, and that this transformation is dependent on Piezo1 activation.

Finally, we asked if the observed bimodal presentation of NF-κB activation (Fig. 4A) is indeed related to mechanical signal from the scar matrix in PAS patients. We co-analyzed Piezo1 and RelA localization in decidual fibroblasts using immunohistochemistry after verifying the specificity of Piezo1 antibody (Supplementary Fig. 5C, D). Regions with mostly nuclear RelA localization also showed higher Piezo1 expression (Fig. 4N). To note, in vitro Scar matrix had also resulted in a similar twofold change in expression for *PIEZO1* mRNA (Fig. 4B), consistent with PAS in vivo scRNAseq data (Supplementary Fig. 5B). These results are noteworthy as it shows that chronic mechanical stimuli from scar can even change expression of Piezo1. Utilizing scRNAseq data from normal pregnancy[59,60], and PAS patients[50], we also found coexpression of *PIEZO1* with *CXCL8* (Supplementary Fig. 5K, L). Together with increased expression, and higher activity of Piezo1, aberrant matrix on scar results in activation of NF-κB signaling in decidual fibroblasts.

## Scar matrix increases contractile force generation and membrane tension in decidual fibroblasts driving Piezo1 activation

We asked how Scar matrix activates Piezo1. Uterine scar presents a composite chemo-mechanical stimulus to the ESFs decidualizing in its proximity. These include higher rigidity, high collagen content, as well as reduced isotropy in the ultrastructural arrangement of collagen fibrils. Indeed, Yap, a key mechanical transcriptional regulator active in cells on rigid surfaces, was mostly cytoplasmic in decidual cells distal to the acellular scar in PAS tissue. In contrast, Yap was mostly nuclear in the fibroblasts proximal to scar (Fig. 5A and Supplementary Fig. 7A). In vitro ESFs decidualized on Scar and Physio matrix also showed a similar Yap localization (Fig. 5B and Supplementary Fig. 7B). Moreover, we found elevated *YAP1* expression in adherent decidua of PAS patients (Supplementary Fig. 7C). Furthermore, elevation of nuclear Yap expression led to increased Piezo1 expression in dESFs (Supplementary Fig. 7D). Several gene ontologies related to cellular contractility, actomyosin organization, and cell-matrix adhesion were significantly upregulated in Scar vs Physio (Supplementary Fig. 2 and 7E). Focusing on gene sets related to cellular contractile

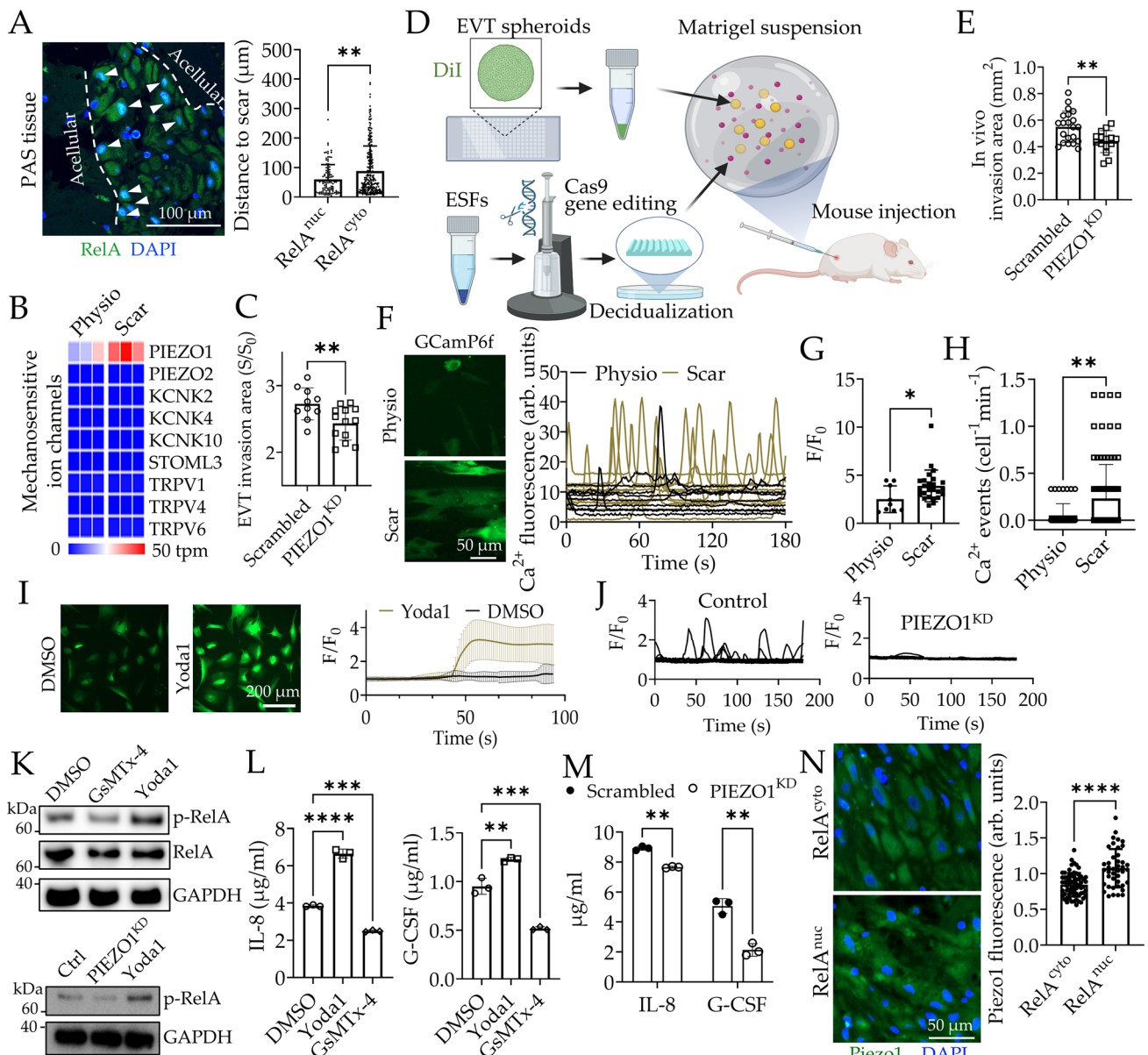

**Fig. 4 | Piezo1 dependent decidual mechanoregulation drives IL-8/G-CSF production. A** Immunohistochemistry images of RelA expression in decidual fibroblasts from PAS patient; Quantification showing distance from scar for dESFs classified according to RelA localization; $n = 104$ and 253 cells. $p = 0.002$.
**B** Heatmap showing tpm values of genes encoding mechanosensitive ion channels in dESFs on Physio or Scar; n = 3 biological replicates. **C** ANSIA analysis of primary EVTs spheroids invasion into scrambled and PIEZO1$^{KD}$ dESFs; $n = 10$ and 13 spheroids. $p = 0.0099$. The scrambled control is shared with Fig. 2F since these conditions are studied in the same round of experiment. **D** Schematic showing HTR8 spheroids invasion into wildtype and gene edited dESFs embedded in Matrigel plugs in mouse. **E** Invasion area of EVT spheroids in Matrigel plugs containing scrambled and PIEZO1$^{KD}$ dESFs. $n = 21$ and 15 spheroids. $p = 0.004$.
**F** Snapshot and calcium transients of dESFs transduced with GCamP6f on Physio and Scar. **G** Ca$^{2+}$ peak/basal ratio and transient events (**H**) in dESFs on Physio and Scar. and $n = 44$ and 92 cells (**G**); $n = 9$ and 25 cells (**H**) $p = 0.03$ (**G**) and 0.002 (**H**).
**I** Images of dESFs loaded with Fluo4-AM treated with DMSO or Yoda1; Graph showing Ca$^{2+}$ peak/basal levels. $n = 20$ cells. **J** Ca$^{2+}$ dynamics in scrambled and PIEZO1$^{KD}$ dESFs. $n = 48$ and 32 cells. **K** Immunoblots showing abundance of RelA, and phosphorylated RelA in dESFs treated with GsMTx-4, or Yoda1, and in PIEZO1$^{KD}$ dESFs. $n = 2$ biologically independent experiments. **L** ELISA measurement of IL-8 and G-CSF concentration in supernatant of dESFs treated with Yoda1 or GsMTx-4. $n = 3$ biological replicates. $p = 4 \times 10^{-6}$, 0.0005, 0.005, and 0.0009.
**M** Concentration of IL-8 and G-CSF in supernatant of scrambled and PIEZO1$^{KD}$ dESFs. $n = 3$ biological replicates. $p = 0.003$ and 0.004. **N** Immunohistochemistry images and graph showing Piezo1 expression in decidual fibroblasts with cytoplasmic or nuclear RelA localization in PAS patients. $n = 42$ and 66. $p = 2 \times 10^{-7}$. Data in all bar graphs are showing as mean ± s.d.; statistical significance are determined by unpaired two-tailed $t$-test (*$p < 0.05$, **$p < 0.01$, ***$p < 0.001$, and ****$p < 0.0001$). Source data are provided as a Source Data file. Figure 4D created with BioRender.com released under a Creative Commons Attribution-NonCommercial-NoDerivs 4.0 International license (https://creativecommons.org/licenses/by-nc-nd/4.0/deed.en).

machinery, we found key ontologies upregulated in Scar vs Physio, including focal adhesion, stress fiber assembly, contractile actin fiber bundle etc. (Fig. 5C and Supplementary Fig. 7F). It is now well established, including in some of our previous works, that both stiffness and anisotropic arrangement of the extracellular matrix can promote the

intracellular actomyosin assembly, driving its maturation, and increasing cellular contractility[61]. Immunostaining indeed revealed highly abundant, organized, parallel bundles of F-actin on Scar vs Physio, with higher mean lengths indicating maturated stress fibers (Fig. 5D).

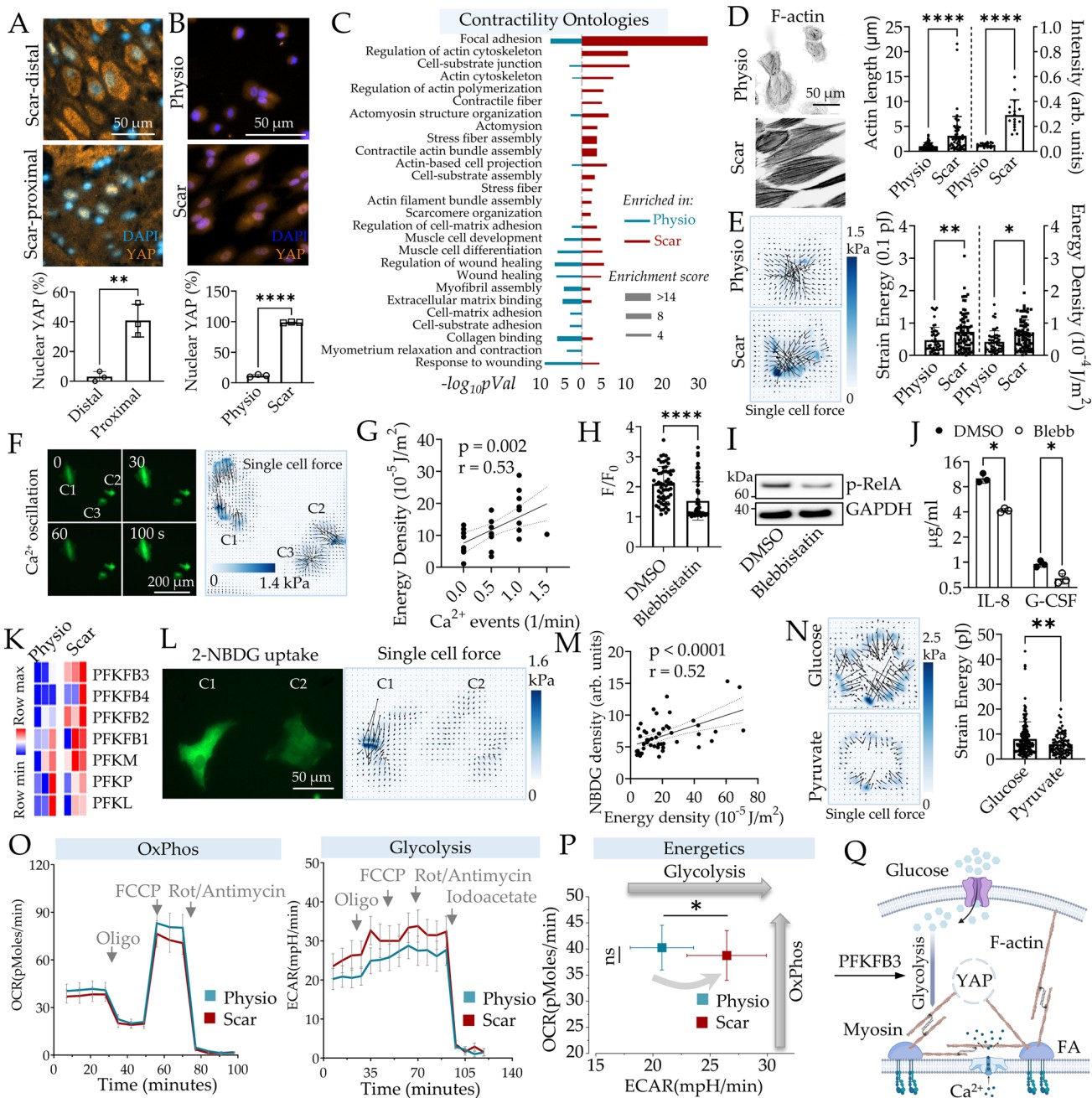

**Fig. 5 | Scar activates Piezo1 through glycolysis fueled actomyosin contraction.**
**A** Immunohistochemistry images of Yap localization in decidual sections proximal, or distal to scar from PAS patients; $n = 3$ sections; $p = 0.005$. **B** Immunofluorescence and graph showing Yap localization in dESFs on Physio and Scar; $n = 3$ biological replicates. $p = 2 \times 10^{-7}$. **C** RNAseq-based enrichment analysis of contractility ontologies in dESFs. **D** F-actin staining in dESFs on Physio, and Scar (left), $n = 61$ cells for each condition; $p = 6 \times 10^{-5}$. Graph showing length and intensity of F-actin bundles (right), $n = 21$ and 17 cells. $p = 9 \times 10^{-11}$. **E** Traction force map of dESFs on Physio and Scar (left). Quantification of strain energy and energy density for each cell (right); $n = 50$ and 82 cells; $p = 0.005$ and 0.01. **F** Time-lapse images showing Ca²⁺ dynamics in three individual dESFs: C1, C2, and C3; heatmap showing corresponding traction force of each cell; **G** Correlation analysis of energy density with Ca²⁺ events frequency. $r$: Pearson correlation coefficient; $n = 27$ cells. **H** Ca²⁺ events and abundance of phosphorylated RelA (**I**) in dESFs treated with DMSO or Blebbistatin; $n = 60$ and 59 cells; $p = 2 \times 10^{-6}$. **J** Concentration of IL-8 and G-CSF in supernatant of dESFs treated DMSO or Blebbistatin by ELISA; $n = 3$ biological samples; $p = 0.002$ and 0.01. **K** Heatmap showing gene expression of PFKs and

PFKBs in dESFs. **L** Images of two dESF cells showing 2-NBDG uptake (left) and their co-measured traction force maps (right). $n = 50$ biological replicates. **M** Pearson correlation analysis of cellular energy density and mean 2-NBDG intensity in dESFs; $n = 50$ cells. **N** Traction force maps and strain energy of dESFs maintained in glucose and pyruvate withmatching C molarity; $n = 164$ and 94 cells; $p = 0.006$. **O** Energetics profiling of oxygen consumption rate (OCR), and extracellular acidification rate (ECAR) in dESFs at basal levels, coupled (oligomycin sensitive), and uncoupled (FCCP sensitive) respiration. $n = 3$ biological replicates. **P** dESFs on Scar show significant increase in glycolysis. $n = 3$ biological replicates; $p = 0.03$.
**Q** Schematic showing Scar promoted cellular contractility is fueled by increased glycolysis. Data in bar graphs are showing as mean ± s.d.; statistical significance is determined by unpaired two-tailed t-test (*$p < 0.05$, **$p < 0.01$, and ****$p < 0.0001$; ns not significant). Source data are provided as a Source Data file. **Q** created with BioRender.com released under a Creative Commons Attribution-NonCommercial-NoDerivs 4.0 International license (https://creativecommons.org/licenses/by-nc-nd/4.0/deed.en).

To functionally confirm increased coupling of Scar matrix with the intracellular actomyosin assembly, we used traction force microscopy (TFM) on ESFs which were decidualized on Scar and Physio, and thereafter replated on TFM substrates of 6 kPa rigidity matching normal endometrium (Fig. 1E). A planar monolayer of fluorescent beads is embedded in a pliable polyacrylamide hydrogel conjugated with Collagen type I. As cells form focal adhesion with the matrix, intracellular actomyosin-generated contractile forces are transmitted to the gel as traction force, resulting in a measurable displacement of fluorescent beads, used to back-calculate strain energy of the cell[62]. We found that Scar resulted in significant increase in dESFs' contractility both at per cell level (strain energy), as well as spatially (energy density) (Fig. 5E).

ESFs decidualized on Scar showed higher Piezo1 activation, resulting in increased NF-κB mediated inflammatory transformation. We therefore asked if the intracellular mechanical changes within dESFs on Scar could contribute to Piezo1 activation. We co-measured $Ca^{2+}$ oscillations in a population of dESFs along with their traction forces using TFM at a single-cell level. Harnessing the population-wide co-variance of either quantity, we found that frequency of $Ca^{2+}$ oscillations was correlated with cellular strain energy (Fig. 5F, G, Supplementary Fig. 7G, and Supplementary Movie 10). Addition of Blebbistatin, a myosin II inhibitor expected to reduce contractile force generation, resulted in significant reduction in $Ca^{2+}$ dynamics (Fig. 5H, Supplementary Fig. 7H, and Supplementary Movie 11). Finally, we also found that Blebbstatin treatment decreased phospho-RelA abundance, as well as IL-8 and G-CSF production, measured by ELISA (Fig. 5I, J). Overall, these data showed that increased contractile force on Scar activated Piezo1-mediated $Ca^{2+}$ signaling, contributing to increased IL-8/G-CSF production.

## Scar induced enhancement in fibroblast traction force generation is accompanied with increased reliance on glycolytic metabolism

Scar matrix resulted in increased contractile force generation in dESFs, which requires maturation of the actomyosin assembly (Fig. 5D). High cellular contractility necessitates increased energy utilization. Differential gene expression analysis showed that genes encoding isoforms of 6-phosphofructo-2-kinase/fructose-2,6-bisphosphatase (*PFKFB*s) were upregulated on Scar (Fig. 5K). *PFKFB3* is of particular interest among the four isozymes due to its highest capability of promoting glycolytic flux and keeping glycolysis high. It's frequently overexpressed in numerous human tumors, including ovarian, lung, breast, colon, pancreatic, and thyroid tumors[63,64]. To test the relationship between glucose uptake and cellular contractility, we harnessed the population-wide variation between either quantity in single cells and measured their correlations. A strong correlation existed between glucose uptake and strain energy, suggesting that dESFs primarily require glycolysis for contractile force generation (Fig. 5L, M and Supplementary Fig. 7I). To explore the link between NF-κB pathway to glycolysis, we knocked down RelA expression in dESFs and performed 2-NBDG uptake assay. We found that RelA knockdown reduced 2-NBDG uptake in dESFs (Supplementary Fig. 7J, K), indicating that NF-κB pathway and glycolysis are correlated. Culture in equimolar (C content normalized) pyruvate significantly decreased strain energy, suggesting that a shift to the citric acid cycle reduces contractile force generation on Scar (Fig. 5N). Recent report indicated that F-actin bundling sequesters E3 ligase TRIM21, which modulates the degradation of rate-limiting metabolic enzyme phosphofructokinase, thereby coupling actin polymerization with glycolytic flux[65]. To document the energetic state of dESFs on Scar, we tested the oxygen consumption rate (OCR) and extracellular acid reflux rate (ECAR) using Seahorse XFe metabolic analyzer. Although we did not find any significant difference in OCR rates between dESFs previously cultured on Physio and Scar, glycolysis was significantly increased on Scar matrix (Fig. 5O, P). These data suggested that increased energy demand of the Scar transformed dESFs is primarily met by increased glycolysis (Fig. 5Q).

## Protein Kinase C (PKC) mediates increased $Ca^{2+}$ signaling related inflammatory transformation of dESFs on Scar

We sought to identify the potential signaling intermediaries regulating Ca2+-mediated NF-κB activation in dESFs on Scar. There are multiple cellular sensors of the cytosolic $Ca^{2+}$ levels, including calmodulin, phosphatidylinositol 3-kinase (PI3K)/Protein kinase B pathway, and Protein kinase Cs (PKC), all of which have been previously shown to regulate NF-κB activation[66]. Kinase enrichment analysis identified *AKT1, PRKCA, EKR1/2* as top enriched kinases in dESFs from Scar (Fig. 6A). We therefore tested if PKC was activated in dESFs on Scar using immunoblot for PKC substrates. Indeed, we found increased signal for PKC-mediated phosphorylation on Scar (Fig. 6B). Next, we ask if the PKC activation is Piezo1 dependent by treating dESFs with GsMTx-4 (Piezo1 inhibitor), Yoda1 (selective Piezo1 activator), or Gö6983 (PKC inhibitor). We found that GsMTx-4 reduced phosphorylation of PKC substrates, while Yoda1 increased it, confirming that PKC activation in dESFs is Piezo1 dependent. (Fig. 6C and Supplementary Fig. 8). We then asked if NF-κB phosphorylation on Scar could be explained by PKC activation. Gö6983 reduced abundance of phospho-RelA (Fig. 6D). Furthermore, Gö6983 reversed the increased phospho-RelA abundance achieved by Yoda1 (Fig. 6E), together suggesting that Piezo1 mediated increased $Ca^{2+}$ signaling activated PKC, which phosphorylated NF-κB. Could PKC activation therefore contribute to inflammatory cytokine production on Scar? To test this, we perturbed PKC activity in dESFs on Scar, and used ELISA to measure IL-8/G-CSF, cytokines responsible for recruiting EVTs towards the Scar-decidua. For both IL-8/G-CSF, we observed that Gö6983 significantly reduced production, while PKC activator, PMA increased it significantly more than control (Fig. 6F). Finally, we tested if PKC is a key intermediate second messenger for Piezo1-mediated IL-8/G-CSF production. ELISA confirmed that while Yoda1 increased IL-8/G-CSF production in dESFs as previously noted, addition of Gö6983 along with Yoda1 reversed the increase (Fig. 6G). These data showed that opening of Piezo1 mechanosensory channel on Scar resulted in increased $Ca^{2+}$ activity which activated NF-κB via PKC signaling.

## MafG stabilization by Piezo1 regulates transcription of trophoblast recruiting cytokines

Predicted transcription factor activation on Scar also showed several zinc finger nucleases, *PAX3*, and *MAFG* (Fig. 7A). MAFG is enriched in endometrium and placenta, but very little is known about its role in pregnancy. Small Maf factors seem to play a role in regulating transcription of inflammatory cytokines, with a report in the myometrium[67], and another in the central nervous system[68]. MAFG transcripts level was significantly increased in Scar (Fig. 7B), as well as MafG abundance (Fig. 7C). Consistently, we also found increased *MAFG* expression in adherent decidua of PAS patients[50] (Fig. 7D). Immunoblot showed that MafG abundance was directly regulated by Piezo1-mediated signaling. While addition of Yoda1 dramatically increased MafG levels within an hour of addition, indicating of post translational regulation, addition of Piezo1 inhibitor GsMTx-4 decreased MafG abundance (Fig. 7E). Interestingly, presence of PKC inhibitor Gö6983 abrogated Yoda1 mediated increase in MafG levels (Fig. 7E). Moreover, inhibition of ERK1/2, a known PKC target, also reduced MafG expression in a dose dependent manner (Fig. 7F and Supplementary Fig. 9A).

We therefore asked if increased MafG levels in Scar could contribute to increased EVT recruitment by the transformed decidual fibroblasts. When *MAFG* gene was silenced in dESFs, there was a significant reduction in IL-8 and G-CSF production, which could not be rescued by Yoda1, highlighting the essential role of MafG in regulating Piezo1-triggered inflammation (Fig. 7G, H). In cell migration assay, HTR8 displacement decreased significantly in conditioned medium from dESFs silenced for *MAFG* vs control dESFs, which was rescued by addition of recombinant G-CSF (Fig. 7I, Supplementary Fig. 9B, and Supplementary Movie 12). Indeed, ANSIA based quantification of

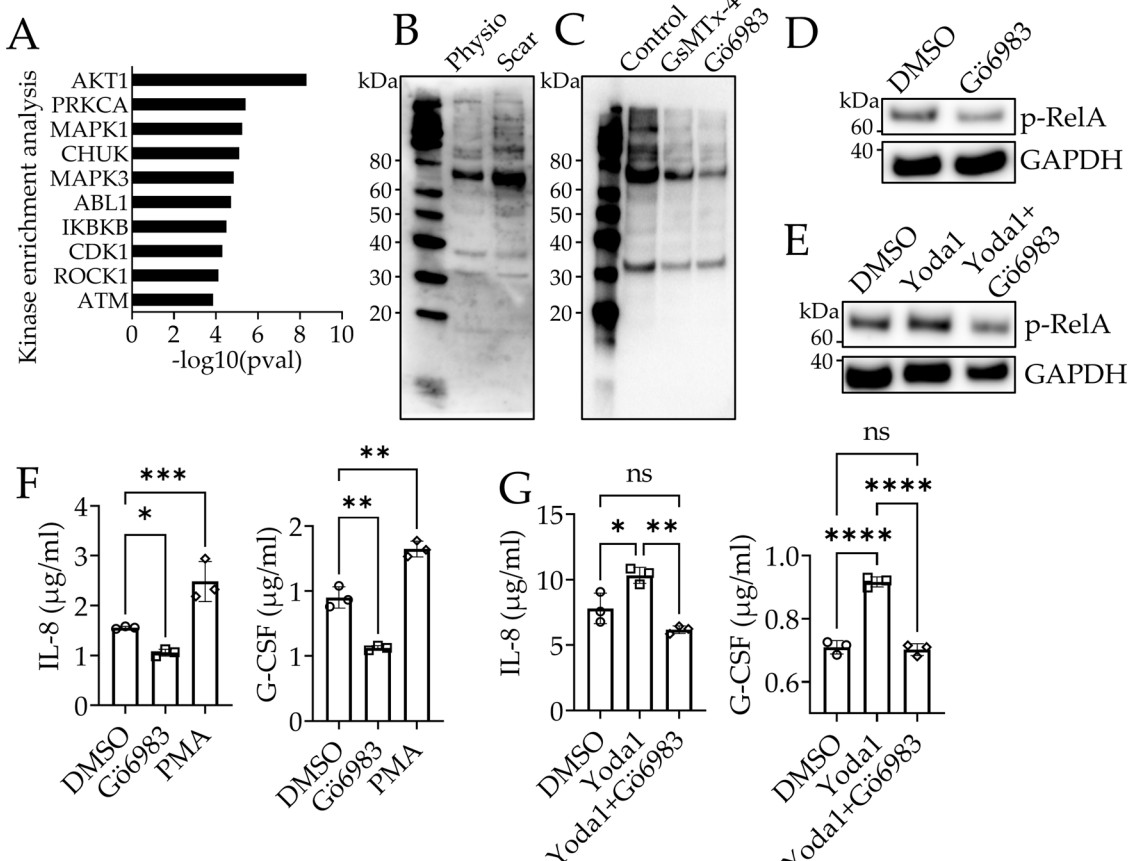

**Fig. 6 | Piezo1-mediated inflammatory transformation of dESFs depends on Protein kinase C (PKC) activation. A** Kinase enrichment analysis (KEA3) predicted top activated kinases in Scar vs Physio dESFs using Fisher's Exact Tests (pval) on RNAseq data. **B** Substrate immunoblot of PKC activated targets in ESFs decidualized on Physio and Scar matrices, as well as (**C**) in dESFs on Scar treated with 4 μM GsMTx-4 and 2 μM PKC inhibitor Gö6983 for 4 h. **D** Immunoblot showing abundance of phosphorylated RelA (p65) in dESFs on Scar without, and after overnight treatment with 2 μM Gö6983, as well as (**E**) 10 nM PKC activator Phorbol 12-myristate 13-acetate (PMA), 5 μM Piezo1 activator Yoda1, and 3 μM Yoda1 plus 5 μM Gö6983; GAPDH is loading control in **D**, **E**. Experiments are one of the two

biological replicates with similar results. **F** ELISA based measurement of IL-8 and G-CSF concentrations in supernatant of dESFs on Scar after overnight treatment with 2 μM Gö6983 and 10 nM PMA; n = 3 replicates; $p = 0.01, 0.0003, 0.001$, and 0.003. **G** ELISA-based measurement of IL-8 and G-CSF concentrations in supernatant of dESFs on Scar after treatment with 3 μM Yoda1, and 3 μM Yoda1 plus 5 μM Gö6983; $n = 3$ replicates; $p = 0.02, 0.002, 9 \times 10^{-6}$, and $8 \times 10^{-6}$. Data in **F** and **G** are showing as mean ± s.d.; statistical significance is determined by unpaired two-tailed $t$-test (*$p < 0.05$, **$p < 0.01$, ***$p < 0.001$, and ****$p < 0.0001$; ns not significant). Source data are provided as a Source Data file.

stromal resistance to primary EVT (Fig. 7J) and HTR8 (Supplementary Fig. 9C, D) invasion showed a similar trend, with dESFs silenced for *MAFG* gene displaying reduced invasibility. Similarly, our Matrigel plug based EVT spheroids invasion of *MAFG*[KD] dESFs in mouse showed the same trend (Fig. 7K, Supplementary Figure 9E). Overall, these data suggest key role of calcium driven PKC signaling, as well as MafG-mediated transcription in regulating the inflammatory transformation of decidual fibroblasts on Scar, promoting chemotactically driven EVT invasion.

## Discussion

The pathogenesis of scarring has long been a research interest since early studies in development and tissue morphogenesis. Therefore, the mechanistic details of the molecular drivers of scarring have been identified. However, the long-term effect of scar on tissue homeostasis is less well documented. Existing scars can profoundly and chronically influence cellular functions in various tissues. Pre-existent scars due to myocardial infarction can increase the chance of heart failure[69,70], and fibrotic response to a malignant neoplasm can regulate cancer dissemination[71,72]. Although there are many case reports of scar-associated invasive pathologies, epidemiological studies are few, and the mechanistic understanding of scar induced

pathogenesis is severely lacking. In a small number of patients burn scars can result in development of squamous cell carcinoma, called Marjolin's ulcers[73,74]. Malignant degeneration of scars have been noted in other cases[75]. Keloids, benign dermal fibro-proliferative tumors with a genetic basis, are positively correlated with several skin cancers and even pancreatic cancers[76]. Mechanistic studies are unavailable because association of scar with invasive processes is small, and takes many years to emerge. Potential emergence of neoplasm is also a relatively smaller concern with scars which typically emerge from significant trauma.

Here, we sought to understand how scar may dysregulate tissue homeostasis by studying a common, surgically induced deep scar, which has a causative pathology: the uterine scar resulting from cesarean procedures which is positively associated with PAS. Using an in vitro model of uterine scar with patient-derived decidua, we showed that scar-like matrix promotes stromal transformation through mechanosensation-mediated inflammation. Scar impairs the physiological endometrial decidualization process, leading to production of chemoattractive cytokines which recruit the EVTs towards the scar. Specifically, the altered mechanical milieu of the scar drives the surrounding stromal fibroblasts to produce IL-8, and G-CSF, which depends on glycolysis-fueled contraction and Piezo1-mediated

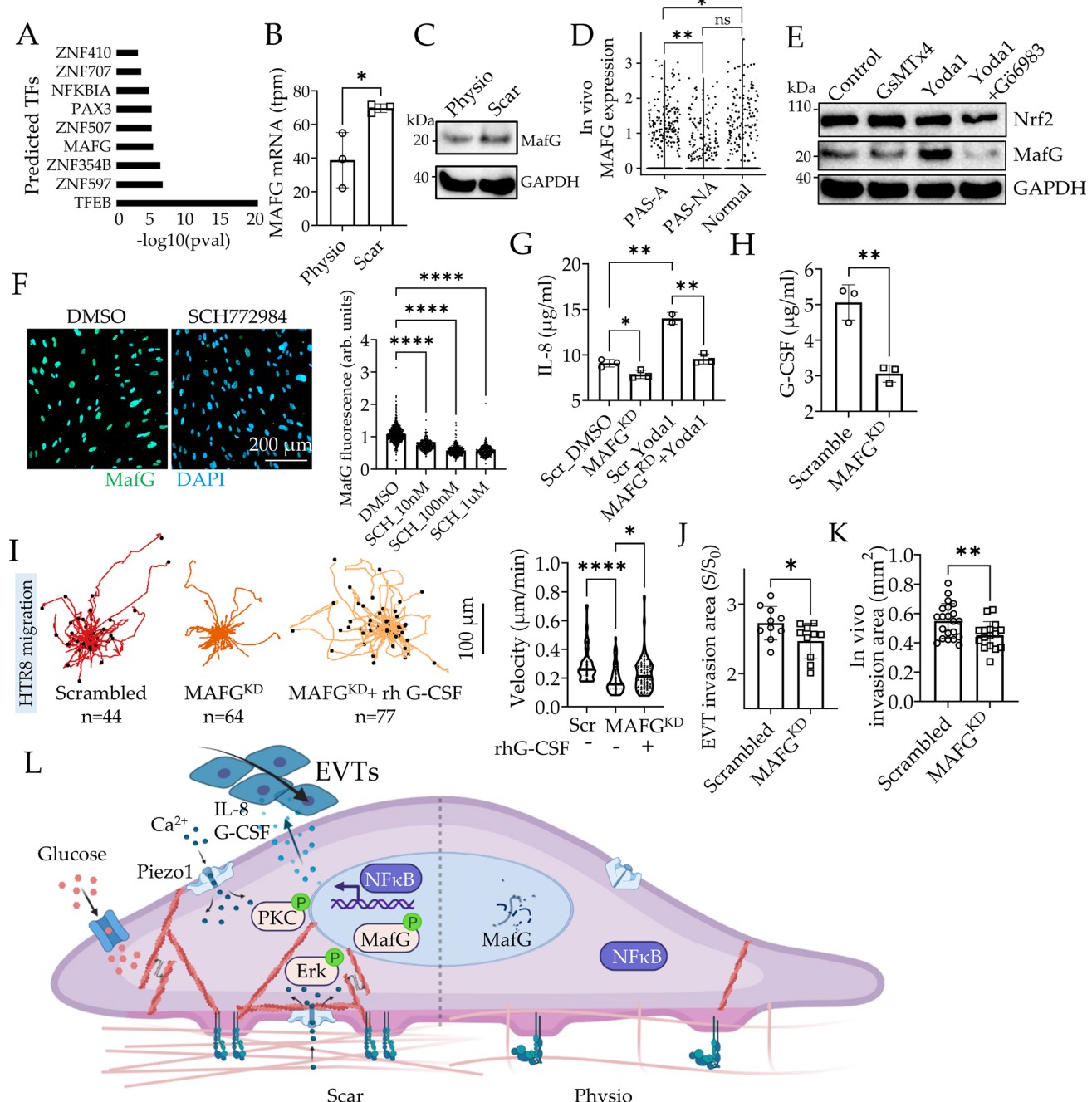

**Fig. 7 | MafG stabilization by Piezo1 mediates transcription of trophoblast recruiting cytokines. A** Prediction of upstream transcription factors on Scar vs Physio dESFs. **B** TPM values of MAFG on Physio and Scar; $n = 3$ replicates; $p = 0.03$. **C** Immunoblot showing MafG abundance in dESFs on Physio and Scar matrices. **D** MAFG scRNAseq expression levels in adherent and non-adherent PAS decidua, and in normal decidua (GEO accession number: GSE212505)[50]. $n = 1836, 1595$, and 1246 cells. $p = 0.04$ and 0.001. **E** Immunoblot showing abundance of MafG and Nrf2 in dESFs on Scar treated with GsMTx4, Yoda1, and Yoda1 plus Gö6983 for 1 h. $n = 2$ biological replicates. **F** Representative immunofluorescence image of dESFs treated with DMSO, or SCH772984; Graph showing quantification of MafG levels in dESFs treated with SCH772984 (SCH) (right). $n = 599, 323, 384$, and 331 cells for each condition; $p = 4 \times 10^{-298}, 0$, and $4 \times 10^{-140}$. **G** IL-8 levels in supernatants from scrambled and MAFG[KD] dESFs treated with Yoda1 by ELISA; $n = 3$ replicates; $p = 0.03, 0.002$, and 0.004. **H** G-CSF levels in supernatants from scrambled and MAFG[KD] dESFs; $n = 3$ replicates; $p = 40.003$. **I** Migration trajectories and mean velocities of HTR8 conditioned with medium from scrambled and MAFG[KD] dESFs,

without, or with addition of 300 ng/mL rh G-CSF. $p = 1 \times 10^{-6}$ and 0.03. **J** ANSIA-based analysis of primary EVTs invasion into scrambled and MAFG[KD] dESFs, without, or with addition of 300 ng/mL rh G-CSF; $n = 10$ spheroids for each condition; $p = 0.03$. The scrambled control is shared with Fig. 2F and Fig. 4C since these independent conditions are studied in the same round of experiment. **K** Invasion area of EVT spheroids in Matrigel plugs containing scrambled and MAFG[KD] dESFs. $n = 16$ and 21 spheroids, respectively. $p = 0.0096$. The scrambled control is shared with Fig. 4E since these conditions are studied in the same round of mouse injection. **L** Schematic showing the plausible mechanism driving inflammatory transformation of decidual fibroblasts and EVT recruitment proximal to existent uterine scar. Data in all bar graphs are shown as mean ± s.d.; Statistical significance is determined by unpaired two-tailed $t$-test (*$p < 0.05$, **$p < 0.01$, and ****$p < 0.0001$; ns not significant). Source data are provided as a Source Data file. **L** created with BioRender.com released under a Creative Commons Attribution-NonCommercial-NoDerivs 4.0 International license (https://creativecommons.org/licenses/by-nc-nd/4.0/deed.en).

calcium signaling resulted NF-κB p65 phosphorylation and MafG stabilization (Fig. 7L).

Our findings differ from the currently held hypothesis explaining PAS pathogenesis, which posits that the acellular scar presents an empty pavement for the EVTs to rapidly invade into, resulting in manifestation of PAS. A counterargument to the hypothesis is that any acellular scar region which is penetrable for EVTs is also by definition, penetrable to numerous other maternal cell types, including the immune cells. Therefore, if scar matrix is merely a spatial void which gets filled in post implantation, it should have been re-celluarized before implantation. Indeed, in patient derived term tissues with PAS, we frequently found acellular, highly collagenous regions, which were not invaded by maternal or placental cells. Remarkably, residual decidua proximal to these acellular scar regions contained more HLA-G⁺ EVTs compared to distal decidua. Confirming our hypothesis even more was the observation that decidual regions proximal to scar contained maternal decidual cells with nuclear RelA, and also expressed higher Piezo1 levels than distal regions. We therefore posit that the altered mechanical cues presented by the scar matrix transform the fibroblasts into an inflammatory state, which then promote EVT recruitment via chemotactic cues. It is possible that once EVTs are recruited more than optimally proximal to the scar, other mechanisms may promote deeper invasion, some of which our study also hints at, including increased contractile force generation by the fibroblasts at the scar.

In recent years, molecular analysis of PAS samples has revealed markers characterizing, or priming aggressive trophoblast invasion[77,78]. Our study details a mechanistic link from the non-physiological mechanical cues present at the scar to the altered intracellular signaling, resulting in PAS. In particular is the role of a key stretch-gated $Ca^{2+}$ permeable channel, Piezo1, which we found to be both activated as well as increased in expression on Scar. Increased $Ca^{2+}$ signaling activated PKC mediated NF-κB phosphorylation, and transcription of inflammation related genes. Among these inflammatory cytokines, we showed that IL-8, and G-CSF, which significantly increase in decidual Scar, are potent recruiters of EVTs towards the Scar, presenting attractive targets to modulate placental invasion. Piezo1 activation on decidual scar is particularly notable, as it has been described to be important in mechanosensation in cells with large ranges of stretch stimuli, e.g. lung alveolar cells[79]. Further downstream, PKC activation has been previously described to interfere with decidualization[80]. We show that PKC-driven activation of NF-κB, which is well documented in other tissues[81], can play a critical role in transformation of the decidual cells into inflammatory fibroblasts.

We also found a hitherto less studied transcription co-regulator, MafG, as a key contributor to the PAS-like phenotype. Small Maf proteins act as bZIP (basic region leucine zipper) type transcription factors which can bind to DNA, and are well known to form heterodimers with other CNC (cap and collar) transcription factors, which include the antioxidant Nrf (1,2,3) factors[82]. Recently there have been reports of role of MafG in the inflammation of the central nerve system, as well as in cancer[68,83]. We found that scar-like matrix resulted in increased expression and abundance of MafG, driven by Piezo1 activation, suggesting mechanoregulation of MafG activity. The link from cellular mechanosensation to MafG mediated inflammatory response may be more general in other mechanically active tissues.

PAS is a rapidly growing concern during pregnancy, with devastating effects on maternal health and future reproductive plans. We used this model of scar owing to its standardized presentation in many women, which has also resulted in a clearly identifiable pathology caused by the scarring. However, deep scars resulting from trauma to other tissues are also common, and similar scar-mediated inflammatory transformation of the resident fibroblasts can contribute to pathology. Our work presents a mechanistic understanding of the long-term scar-induced effect on fibroblast phenotype, as well as avenues to prevent fibroblast inflammation, a harbinger of various chronic pathologies.

## Methods

### Tissue collection, cell isolation and culture

Human endometrium tissues and FFPE tissue samples from medically necessary biopsies or hysterectomies were obtained from the biorepository at University of Connecticut Health Center after de-identification in accordance with the guidelines and Accio Biobank Online with IRB exemption (Supplementary Table 1). Endometrial tissue was minced on the day of collection, and then either plated on Physio or Scar matrices, or on polystyrene culture plates for isolation of endometrial stromal fibroblasts (ESFs) by selective attachment[84]. Isolated ESFs were maintained in phenol red free DMEM/F12 50:50 containing 25 mM glucose, and supplemented with 10% charcoal-stripped fetal bovine serum (Thermo Fisher), 1% antibiotic/antimycotic, and ITS (insulin, transferrin, and selenium). To induce decidualization, ESFs were treated with 0.5 mM 8-bromo-cyclic 3′,5′-(hydrogen phosphate)-adenosine (Cayman Chemicals) and 1 μM Medroxyprogesterone acetate (Cayman Chemicals) in DMEM/F12 medium with 2% FBS for 4 days. Chemical perturbations are implemented during the last day of decidualization unless otherwise stated. For 3D decidualization, cells seeded on Physio and Scar matrices were embedded in 1 mg/ml type I collagen from rat tail (Thermo Fisher) and incubated at 37 °C for 1 h to allow 1 mm thick collagen gel formation. Decidualization medium was then added without disturbing the matrices.

To isolate HLA-G⁺ cells, decidual basalis was separated from the maternal side of the termed placenta tissue and then minced. Enzyme cocktail containing dispase, collagenase I, and DNase I was added to equal volume of minced tissue. The mixture was fixed on a shaker with 500 rpm at 37 °C for 1 h. Equal volume of culture medium containing 20% FBS was added to the mixture to block the digestion. The mixture was then centrifuged at 340 × $g$ for 5 s to remove the undigested tissue. The supernatants containing isolated cells were collected and directed plated into flasks coated with 20 μg/ml fibronectin. After 48 h of culture, non-attached cells were gently washed away. The isolated placental cells were then detached and blocked with 5% goat serum and 1% BSA at room temperature for 20 min, before stained with Alex Flour 488 conjugated human anit-HLA-G antibody (Biolegend, 1:50) at 4 °C for 30 min. HLA-G+ cells were sorted using flow cytometry (BD FACSAria II sorter) and maintained in DMEM/F12 with 10% FBS, ITS, and antibiotics.

Human extravillous trophoblast cell line-HTR8/SVneo derived from first trimester of pregnancy were obtained from ATCC (CRL-3271). Cells were cultured in RPMI medium supplemented with 10% FBS and 1% antibiotic/antimycotic (Gibco). HTR8 were stably transduced with plasmid expressing H2B-bound mCherry driven by CMV promoter unless otherwise stated.

### Mechanical indentation

Endometrial tissue samples from patients were cut into 1 cm × 1 cm × 0.4 cm pieces and half-embedded in low-melting agarose and then immersed in RPMI medium. Mechanical indentation was implemented using Mach-1 micromechanical testing system (Biomomentum) mounted with an indentation probe with a 2 mm bead at the tip. The indentation speed is 50 μm/s and the indentation depth is 500 μm. At least 20 locations were measured per sample.

### Matrix fabrication

Scar matrix was fabricated by sandwiching polyacrylamide precursors between nanotextured poly(urethane acrylate) (PUA) molds and saline-activated coverslips. PUA molds are either pre-fabricated[61], or fabricated from a silicone mold fabricated using electron-beam lithography followed by deep-reactive ion etching. Saline-activated

coverslips for gel attachment were cleaned with ethanol and sonication, treated with air plasma, and activated with 0.5% glutaraldehyde and 0.5% (3-Aminopropyl)triethoxysilane (Sigma Aldrich). Poly-acrylamide precursor solution containing 10% acrylamide and 0.3% bis-acrylamide (Bio-Rad) was degassed for 30 min and mixed with 0.1% tetramethylethylenediamine and 0.1% ammonium persulfate (Sigma Aldrich) before sandwiched between silane-activated coverslips and PUA molds for 20 min. The physiological matrices were fabricated following similar procedures but with precursor containing 5% acrylamide and 0.12% bis-acrylamide sandwiched between saline-activated and Rain-X coated coverslips. The crosslinked gels were peeled off and coated with 40 µg/ml (Physio) and 100 µg/ml (Scar) collagen type I using sulfo-SANPAH (Thermo Fisher) overnight at 4 °C. Gels were sterilized under UV for at least 2 h before cell seeding.

### Collagen orientation analysis
Tissue slides from PAS patients were stained with Picrosirius red. Collagen signals in the regions of interest were obtained after color deconvolution using ImageJ. Collagen orientation was quantified using OrientationJ plugin in ImageJ.

### AFM imaging
AFM imaging of the surface topography of the Scar matrix was performed using Asylum Research Cypher AFM in PBS. A 0.08 N/m tri-angle PNP-TR probe (NanoWorld) was used. The scanning speed was set to 0.3 Hz.

### In vitro invasion assays
For in-situ invasion, HTR8 spheroids prepared using ultra-low attachment round bottom microplate (Corning) were plated on top of dESFs on Physio and Scar substrates. Time-lapse images were taken every 2 h for at least 24 h. Invasion area was normalized by the initial projection area of the spheroid. For spheroid invasion, invading primary EVT or HTR8 spheroids were seeded on dESFs monolayers on nanopatterned scar-mimicking substrates. Time-lapse images were taken, and invasion area was analyzed as aforementioned. For ANSIA invasion, invading primary EVTs or HTR8s and dESFs monolayers were patterned juxtaposed to each other as previously described[24,42]. Briefly, a custom-made polydimethylsiloxane stencil was placed on the nanogrooves-patterned substrate. The device was kept in a vacuum to remove air bubbles under the stencil. Then, HTR8-mcherry were seeded at a density of $5 \times 10^5$ cells and allowed to attach to the substrate overnight. The stencil was removed carefully using blunt-end tweezers. The unlabeled stromal cells were seeded at a density of $5 \times 10^5$ to fill and attach that area covered by the stencil before. The unattached cells were washed off after 5 h of incubation. Invasion were recorded for 24 h. Invading cells were imaged by using time-lapse microcopy every 1 h for 24 h. The area occupied by the invading cells was traced manually using Region of Interest (ROI) panel in the Fiji software. The normalized extent of invasion was calculated by dividing total $\delta Area(t)$ (of invading cells) by the length of the initial trophoblast–stroma interface.

$$\delta Area(t) = Area(t) - Area(t0) \qquad (1)$$

$$\langle \delta Area(t) \rangle = \frac{\delta Area(t)}{L(Interface)} \qquad (2)$$

Automated peak identification was performed on the converted ROIs to a one-dimensional mask. The masks' profiles were smoothed by moving the average over 20 pixels. The mean signal to either side of every point with 40 pixels each, was calculated by using smoothed profiles. A peak also was identified when the boundary of smoothed profile was larger than both side average. Finally, the number of invasive forks as well as the distribution of the depth of invasion in stromal monolayer were measured.

For 3D invasion, ESFs were seeded on Physio and Scar. After cell attachment, 1 mg/ml collagen solution were casted on ESFs and incubated at 37 °C for 1 h to form a gel layer with thickness of 1 mm. After 4 days of decidualization, HTR8 spheroids prepared as aforementioned were suspended in 1 mg/ml collagen solution and plated on the 3D decidualization ESFs and incubated at 37 °C for 1 h to allow spheroid settle down and gel formation. dESF monolayer and HTR8 nuclear locations were recorded by Zeiss Apotome 3D scanning. Distance between each individual HTR8 nucleus and each dESF layer were calculated after deconvolution and nuclear segmentation.

### In vivo invasion
Matrigel plug based EVT spheroid in vivo invasion was performed in mouse. Briefly, target genes in primary ESFs were knockdown using Neon NxT electroporation system (Invitrogen) following the manufacturer's guideline. ESFs were then seeded on scar-like substrate and decidualized for 8 days. One day before injection, growth factor reduced high concentration Matrigel (Corning) was thawed in ice at 4 °C overnight for cell encapsulation. On the same day, HTR8 cells were labeled with 4 µM CM-DiI (Invitrogen) for 5 min at 37 °C and 15 min at 4 °C. Afterwards, HTR8 spheroids were prepared by centrifuging 1000 cells in each well of the 384-well spheroid microplate (Corning) at 350 × g for 5 min. After 24 h, HTR8 spheroids were collected in 1.5 mL tubes (25 spheroids/tube). Meanwhile, dESFs were collected and resuspended in 50 µL culture medium in the same tubes containing HTR8 spheroids (1 million cells/tube). The tubes were then placed in ice. Before injection, 1 mL syringes and 20 G needles were also cooled in ice. Then, 150 µL Matrigel was draw into each syringe without needle. After mounting the capped needle onto syringe, the whole syringe was embedded in ice immediately to prevent Matrigel gelation. Before injection, the back fur of male mice was removed by shaving and cream. After anesthetization, HTR8 spheroids and dESFs in each 1.5 mL tube were mixed with Matrigel pre-loaded in each syringe and injected subcutaneously into the back of the mice. No more than 4 injections were performed for each mouse, with control and experimental conditions in the same mouse. Adult (three months old) SCID/beige male mice (Inotiv) were used in this study to reduce the variations caused by female progesterone on dESFs. 10 mg MPA suspension were also injected subcutaneously into each mouse to help dESFs maintain their differentiation status. At least two injections on two mice were performed for each condition. Injection sites were marked with marker pen to help Matrigel plug excision 3 days after injection. To maintain the integrity of Matrigel plugs, skin tissue was excised together with the plugs. The whole tissue was then fixed in 0.5% Glutaraldehyde (EMS) for at least 2 h to crosslink Matrigel, and then in 4% PFA overnight. Skin tissue was then carefully removed to release Matrigel plugs for imaging. To ensure that the integrity of EVT spheroids is not affected by the forces during mixing and injection, the same procedures were also performed in vitro in 96-well plate instead of in mouse. We confirmed that the integrity of spheroids was not affected, and its capability of invasion remained (Supplementary Fig. 6). All animal protocols were approved by the Institutional Animal Care and Use Committee (IACUC) at the University of Connecticut Health Center before study initiation. All experiments were performed in accordance with IACUC guidelines, and abides by the ARRIVE guidelines for reporting animal experiments. Mice were kept in a 12 h light–dark cycle, temperature-controlled (22 ± 2 °C) and humidity-controlled (55 ± 5%) environment and fed a standard chow diet. Mice were anaesthetized with isoflurane and cells were subcutaneously injected into the mice. After 3 days injection, Mice were euthanized with carbon dioxide and Matrigel plugs, skin tissue were harvested for analysis.

## RNA sequencing and transcriptomic analysis

Cells were lysed and RNA was isolated with RNeasy Mini Kit (Qiagen) following manufacturer's instructions. RNA integrity was evaluated with Bioanalyzer 2100 (Agilent) and RIN ~ 8 was used for library preparation. Library prep and RNA sequencing were performed by Novogene Inc. HISAT2 pipeline was used to align reads to NCBI GRCh38 genome assembly. HTSeq was used for reads were counting, and DESeq2 was used for statistical significance (p-values) and fold-changes for differential expression. Enrichment of gene sets in the differentially expressed (DE) genes were calculated using Fisher exact test to calculate the overrepresentative of terms (Gene Ontology, Hallmark, Wikipathways) using hypergeometric test followed by correction for multiple testing[85]. Ingenuity Pathway Analysis (Qiagen Inc) was used to calculate predicted scores for transcription factors activation and canonical pathways analysis. Gene set enrichment analysis (GSEA) was performed on the genes[86]. Hierarchical clustering was performed using UPGMA method with Euclidian distance on z-scores as mentioned earlier[61]. KEGG pathway analysis was performed using ShinyGO8[87,88]. Kinase enrichment analysis was performed using the webserver application Kinase Enrichment Analysis 3[89]. For single-cell RNA sequencing analysis, decidua cells were extracted from the data following the same annotation as described by the study (GEO accession number: GSE212505)[50]. Decidua cells from PAS patients at the adherent and non-adherent sites, and from normal pregnancy were analyzed. Genes of interest were visualized using violin plots with t-test. Correlation analysis of gene expression was performed using the Spearman method. Pathway enrichment analysis for differential expression genes in decidua cells was performed using GSEA on hallmark pathways.

## Immunoblotting

Cells were harvested and lysed in cell lysis buffer containing RIPA buffer (Bio-Rad), protease and phosphatase inhibitor cocktail (Sigma Aldrich). BCA kit (Pierce) was used for protein concentration measurement and normalization. Denatured samples (70 °C for 10 min in SDS) were loaded on 4–12% NuPAGE Bis-Tris Gel (Thermo Fisher) along with Lamelli loading buffer. Proteins were then transferred to PVDF membranes, and blocked with 3% BSA for 1–2 h at room temperature and incubated with primary antibodies (1:1000) overnight at 4 °C. Membranes were washed with TBST for 5 times and re-blocked with 3% BSA. HRP-linked anti-rabbit or mouse IgG secondary antibodies (GE healthcare; 1:10,000) were cross-linked at room temperature for 1 h, and protein bands visualized using enhanced chemiluminescence reagent (Thermo Fisher) using an Imager (Molecular Biosciences). Antibody details are listed in Supplementary Table 2.

## Immunofluorescence

Cells were fixed in 4% paraformaldehyde for 15 min, permeabilized in 0.1% Triton-X100 for 10 min, and blocked in 1% BSA and 5% goat serum for 1 h at room temperature. Cells were then incubated with primary antibodies at 4 °C overnight and with secondary antibodies for 1 h at room temperature. FFPE sections were rehydrated by immersing the slides in xylenes twice for 3 min, 1:1 xylenes:ethanol for 3 min, 100% ethanol twice for 3 min, followed by 90%, 75%, and 50% ethanol for 3 min each. Antigen retrieval was performed after rehydration by immersing the slides in citrate buffer at 95 °C for 30 min. Slides were then incubated with primary antibodies (1:100) at 4 °C overnight and with secondary antibodies (1:400) for 1 h in the dark at room temperature. Antibody details are listed in Supplementary Table 2.

## Gene silencing

Gene silencing was achieved by using pre-prepared synthetic sgRNA (IDT). ESFs were transfected with sgRNA and TrueCut Cas9 protein (Invitrogen) using Lipofectamine CRISPRMAX transfection reagent (Invitrogen) or Neon NxT Electroporation System (Invitrogen). Specifically, a cocktail was created by mixing (i) solution1: 24 μl Opti-MEM and 1 μL CRISPRMAX, and (ii) solution 2: 10 nmol sgRNA, 15 nmol Cas9, 1.5 μL CRISPRMAX Plus reagent and remaining Opti-MEM to make a 30 μL solution. Solution 1 and 2 were mixed and incubated for 10 min before being drop dispensed into one well of a 24-well plate containing cells at 50% confluency in culture medium. Cells were used after 48 h of transfection, and observation completed within 48 h thereafter. Electroporation based CRISPR/Cas9 gene editing was performed following the manufacturer's guideline for fibroblast gene editing (1650 V/ 20 ms/1 pulse).

## Cell migration and 3D chemotaxis

For cell migration, HTR8-mCherry cells were cultured in conditioned dESF medium mixed with fresh medium (1:1) and monitored using microscope for at least 9 h and time-lapse images were taken every 20 min. Single cell tracking was then performed using Fiji/ImageJ TrackMate plugin. Cell migration velocity and distance were quantified for individual cell. Cell trajectories were plotted using ibidi Chemotaxis and Migration Tool V2.0. 3D Chemotaxis of HTR8-mCherry was performed using μ-Slide Chemotaxis (ibidi) following the manufacturer's instruction. Briefly, $3 \times 10^6$ cells/ml HTR8 were resuspended in 1 mg/ml collagen type I. Six microliters of cell suspension was loaded into the microchannel with a width of 1 mm and a height of 70 μm. For the two chambers separated by the microchannel, one chamber was loaded with 100 ng/ml recombinant human IL-8 or G-CSF (Proteintech), and the other chamber was loaded with vehicle. Cell migration trajectories were quantified as aforementioned. Rayleigh test was performed to determine the statistical significance of chemotaxis. Rayleigh $p < 0.05$ is considered chemotaxis.

## Actin filament analysis

F-actin images were enhanced using the Contrast Limited Adaptive Histogram Equalization (CLAHE) filter, then masks of actin filaments of individual cells were generated by the threshold function in ImageJ. Mean intensity value of actin was then quantified. To obtain actin length, the BoneJ plugin were used.

## Traction force microscopy

Traction force gels were fabricated using protocols previously described[90]. Briefly, coverslips for gel attachment were cleaned with ethanol and sonication, treated with air plasma, and activated with 0.5% glutaraldehyde and 0.5% (3-Aminopropyl)triethoxysilane (Sigma Aldrich). Coverslips for beads coating were treated with air plasma and coated with 0.01% poly-L-lysine (Sigma Aldrich) before coated with carboxylate-modified microspheres with diameters of 0.2 μm (Thermo Fisher). Gel precursor solution containing 7.5% acrylamide and 0.15% bis-acrylamide was degassed for 30 min and mixed with 0.1% tetramethylethylenediamine and 0.1% ammonium persulfate before sandwiched between silane-activated coverslips and bead-coated coverslips for 20 min. After peeling off the bead-coated coverslips, the resulting traction force gels were coated with 30 μg/ml collagen type I using sulfo-SANPAH (Thermo Fisher) overnight at 4 °C. Gels were sterilized under UV for at least 2 h before cell seeding. Images containing microbeads location before and after cells trypsinization were recorded using Zeiss Observer A1 microscope. Traction forces were calculated following protocols previously described[91].

## Glucose uptake

Pre-warmed glucose-free DMEM (Thermo Fisher) with 10% FBS and 400 μM of 2-NBDG (Invitrogen) were added to cells on traction force gel at 37 °C and 5% $CO_2$ for 30 min. Cells were then washed with DMEM twice, added with culture medium, and mounted on microscope for live cell imaging.

## Calcium imaging

To image cellular calcium dynamics, we generated 3rd gen. lentivirus for GCamP6f. Briefly, GCamP6f probe was obtained from Addgene and transferred to pENTR™/SD/D-TOPO® vector (ThermoFischer Scientific). Subsequently, LR reaction was used to transfer it to pLEX_307 vector (Addgene plasmid # 41392). The vector was sequence verified. The virus was generated in HEK293-FT cells using packaging mix of three plasmids (pLP1, pLP2, and pLP/VSVG) from ThermoFisher Scientific. The virus was concentrated using PEG-it™ Virus Precipitation Solution (SBI biosciences) and MOI was calculated using qRT-PCR. Cells were transduced using Polybrene (Millipore Sigma) using MOI 5 (multiplicity of infection). Calcium dyes Fluo4-AM (2 μM, Invitrogen) or Calbryte 590 (5 μM, AAT Bioquest) were premixed with 0.04% Pluronic-F127 (Sigma Aldrich) in serum-free medium and loaded to cells for 20 min. Calcium activities were imaged 20 min after dye washing out. To prevent photobleaching, time-lapse images were taken using light with low intensity and short exposure time at 1 Hz for 2 or 3 min. Photo bleaching of FLuo4-AM was corrected using the Bleaching Correction plugin in ImageJ. To image calcium activities for cells treated with blebbistatin, Calbryte 590-AM (AAT Bioquest) was used as a calcium indicator to avoid autofluorescence of Blebbistatin (Sigma Aldrich).

## Measurement of oxidative phosphorylation and glycolysis

Seahorse XF analyzer (Agilent Technologies) was used to monitor cellular energetics[61]. Cells were dissociated and cultured in a 96 wells XF plate, and 6 replicates were used for each condition. Oxygen consumption rate (OCR) was used to estimate oxidative phosphorylation while change in extracellular acidification rate (ECAR) was used to estimate glycolysis. Basal rates show changes in $O_2$ or pH in the absence of any added compounds or metabolic inhibitors. ATP synthase was inhibited using Oligomycin (4 μM), and rotenone (2 μM) and antimycin A (2 μM) was used to inhibit complex 1/3 respectively. Uncoupling was achieved using FCCP (500 nM) to estimate maximum respiratory capacity, and iodoacetate (100 μM) was used to inhibit glycolysis (glyceraldehyde-3-phosphate dehydrogenase). Respiratory rates were normalized to DNA content using Picogreen DNA assay (Thermo Fisher Scientific) following manufacturer's instructions.

## Enzyme-linked immunoabsorbent assay (ELISA)

Human IL-8 (Biolegend) and G-CSF (SinoBiological) kits were used to test measure secreted IL-8 and G-CSF levels from dESFs culture supernatants according to the manufacturer' instructions. Briefly, a 96-well plate was coated with capture antibody overnight at 4 °C, blocked for 1 h at room temperature, followed by sample incubation for 2 h and detection antibody incubation for 1 h. Then the plate was incubated with Avidin-HRP for 30 min, 10 mg/ml Tetramethylbenzidine for 15 min before 2 N $H_2SO_4$ was added to the plate. The absorbance signals were read using SpectraMax i3x multi-mode microplate reader (Molecular Devices) at 450 nm and background at 570 nm. IL-8 and G-CSF concentrations were quantified using SoftMax Pro.

## Plasmids

pLenti6-H2B-mCherry was obtained from Addgene (#89766). Lentiviral particles prepared with 3rd generation packaging plasmids. HTR8 cells were transduced, and selected using blasticidin resistance, as well as sorted with FACS for reduced heterogeneity of mCherry expression in the cell cultures.

## Microscopy

All cell invasion and migration assays and traction force microscopy were performed using Zeiss Axio Observer Z1 microscope with PECON Incubation System S for live cell imaging. 3D imaging and deconvolution was performed using ZEISS Apotome.2 and Zen Blue 2.6 Pro software. Whole slide scanning was performed using ZEISS Axioscan 7.

## Statistical analyses

Statistical analyses were performed using GraphPad Prism. Two-tailed unpaired Student's $t$-test was implemented for statistical significance of the differences between two groups. One-way ANOVA was performed followed by Tukey test if multiple groups were presented in a graph. For graphs showing correlation analysis, Pearson correlation coefficients were provided together with two-tailed statistical significance values. All bar graphs show mean values with standard deviation as error bars. Statistical significance levels are defined as $*p < 0.05$, $**p < 0.01$, $***p < 0.001$, and $****p < 0.0001$. $P$ values higher than 0.05 are considered non-significant.

## Reporting summary

Further information on research design is available in the Nature Portfolio Reporting Summary linked to this article.

## Data availability

The RNAseq data generated in this study have been deposited in the NCBI GEO database under accession code GSE269722. The publicly available single-cell sequencing data used in this study are available in the GEO database with GEO accession number GSE212505. Source data are provided with this paper.

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

## Acknowledgements
Funding for project was obtained by Startup provided by UCH Department of Biomedical Engineering to Kshitiz, R01 grant by NICHD (1R01HD112424, PI: Kshitiz, M.A.B.), R37 grant by NCI (R37CA248161, PI: Kshitiz) and K99 fellowship by NICHD (K99HD105973, PI: Y.S.). The authors thank Joerg Nikolaus and Yale West Campus Imaging Core for their kind support on AFM imaging.

## Author contributions
W.D. created scar model, performed most experiments, analyzed and interpreted data; A.N. performed invasion experiments; Y.L. performed immunoblots' J.A., Y.S., and S.L. performed bioinformatics and computational analyses; N.R.G., J.R.J., C.M.M., and R.F. provided clinical samples and coordinated with pathology, T.A.S. provided mechanobiological testing, M.S. provided pathological interpretation, and sourced clinical samples; M.A.B. provided clinical interpretation; K. conceptualized the manuscript, supervised the project and wrote the manuscript.

## Competing interests
The authors declare no competing interests.
