## [Peer Review File · Nature Communications]

REVIEWER COMMENTS

Reviewer #1 (Remarks to the Author):

In this manuscript, Du and colleagues develop a uterine in vitro scar model to delineate how mechanical signals play a role into transforming endometrial fibroblasts into scar forming fibroblasts. Through mainly in vitro collagen assays and some complementary tissue placenta accrete spectrum (PAS) specimens, the authors investigate the role of mechanical stimuli in PAS. Although the authors have developed an interesting in vitro assay to study uterine fibrotic fibroblasts, the study lacks rigor overall. The authors claim that invasion of dESFs is critical in determining the aggressive trophoblast invasion without any in vivo assessment and limited human analysis.

Figure 1C: It not clear how the collagen alignment is assessed with the Masson Trichome staining. No obvious difference is shown between the scar distal and scar proximal in the figure provided. Picrosirius red staining would be a better stain to analyze this. Considering the alignment and the elastic modulus are key to in the pattern of the gels this should be more carefully shown.

Figure 1J: The invasion of the two groups shown at 0 and 24 hours does not appear to show a difference. Can the authors provide more convincing data that the invasion is different?

Figure 1L: It is not clear why the authors feel that difference in migration is attributed to cytokine production, without any analysis (...'These data suggested that ESFs decidualized on Scar produce cytokines that recruit HTR8s towards scar')?

Figure 1M: The authors state an increase in inflammatory genes but only CXCL2 is shown?

Figure 1N: The authors state that specific cytokines recruit EVTS preferentially towards the scar but this has not been tested?

Figure 2A: Patient number and demographics are not clearly documented. How many PAS specimens have been included ?

Figure 2G: There is no obvious difference in the RelA expression between scar distal and scar proximal in the data provided in 2G. Can staining for other components of the NFkB transcription factor complex be stained for including NFkBp50 to show a difference?

Figure 3B: The IL-8 staining looks intranuclear and is not clearly different. Why was IL-8 chosen over other cytokines?

Figure 3J: The effect of IL-8 knockdown should assess whether the fibroblasts lose their inflammatory phenotype and not only their migration trajectories. For example, is their fibrotic phenotypes at a molecular level affected?

Figure 4N: It is not clear why Piezo1 was chosen above other mechanical pathways? The link between Yap and piezo 1 was not studied ? It was not clear why the authors evaluated YAP?

Figure 4J: Can the authors explain why there is no difference in RelA expression following GsmTX4 and Yoda, only p-RelA?

Figure 5: The link between NFkB signaling to glycolysis is not clear. Why was PI3K pathway chosen? Immunostaining or RNA scope has not performed? The evidence supporting Piezo1 and PKC activation is not convincing. The difference in the expression between AKT1, PRKCA, EKR1/2 is not clear.

Reviewer #2 (Remarks to the Author):

To authors,

The theme is important. The concept is new and the experimental design was well structured and reasonable. I have minor suggestions.

1. PAS (placenta accreta spectrum; creta (not accreta!), increta, percreta) should be briefly defined regarding its sub-classifications. To tell things strictly, there is not terminology of "placenta accreta" or "accreta". FIGO 2018 effort was mainly to extinguish the ambiguity of the term "placenta accreta", which had two/dual meanings, wide (now, PAS) and narrow (now, placenta creta). Thus, please state/describe "PAS" once you defined classifications, instead of "placenta accreta" or "accreta". In summary, the term "placenta accreta" no longer exists and thus avoid to use it.

2. 1) "PAS pathophysiology" is a "narrower" area compared with "scar pathophysiology" in general. 2) Nature Communications readers are general-medicine > obstetricians. You might, thus, constructed this paper as 1) to determine/identify some aspects of scar physiology, 2) PAS-like model was used, and 3) actually PAS itself is also important pathology recently attracting much wider attention. This is a paper-writing strategy. However, you actually targeted to PAS-model confined to PAS-model); this is "specific" model for PAS. Then, you extrapolated the data to "scar physiology" general. Readers may make question, why PAS to generalize the scar-invasion relationship: for example, A) in the intestinal scar (example; appendectomy scar), the cancer invasion style/manner becomes so different? B) in the episiotomy site, malignant melanoma shows quite (to the extent that comparable to PAS-trophoblast invasion) "different" invasion? There are abundant "parts/ areas" that might indicate the interaction of "scar and invasion". Some scar may not affect invasion at all, whereas PAS-site affect invasion markedly. Putting this "specific" site and then discuss "scar" general; is this scientifically sound? Of course, this study is not to clarify A) or B): this is not your present intention. What I wish to say is; the present structure is OK? First PAS physiology, then relate (speculate/suggest/exploit) this data with other general condition; this may be much "natural", I believe. This is based on my personal experience of 495 paper-writing during 4.5 decades. You studied here "PAS" and not "scar general". I, of course, understand that you CAN speculate/suggest mechanism on scar and cell invasion based on the PAS-model-data. The effort to "change" the paper structure is not difficult: possibly it requires 4-6 hours: only changing the parts before versus after with citation number change. You will hear if you had better incorporate my advice fro EDITOR.

Reviewer #3 (Remarks to the Author):

Du W et al. developed new models to study the pathophysiological manifestations of deep internal scars, e.g., uterine scars due to cesarean surgery. A combination of decidualized endometrial stromal fibroblasts (dESFs) and mechanically distinct tissue scaffolds (Physio Vs. Scar) are used to study the invasion of extravillous trophoblasts (EVTs). Immortalized EVT cells (HTR8/SVneo) are used for assessing trophoblast invasion. Authors show that scar-associated dESFs promote the invasion of HTR8 cells because of increased NFkB signaling and secretion of L-8/G-CSF. This, in turn, is mediated through mechanically activated cation channels, Piezo1. Further evidence demonstrates that glycolysis potentially drives the scar-mediated aberrant responses. PKC is identified as a potential messenger for eliciting scar-induced inflammatory phenotype of dESFs. In addition, MAFG is a suggested factor downstream of Piezo1-mediated activation of EVT recruitment cytokine networks. This comprehensive manuscript addresses an essential question of how internal scars modulate tissue functions. My comments are listed below.

An extensive body of literature shows that Piezo1 is a "cation" channel, not a calcium-selective ion channel. It is stated that "Piezo1 is a membrane stretch-gated Ca²⁺ channel" (Page 7). If Piezo1 channels

in decidualized endometrial stromal fibroblasts (dESFs) selectively conduct calcium, authors should show data on calcium ion selectivity by electrophysiology.

All representative images in the manuscript are composites of two or more channels. Single channel images should be shown along with merged images, particularly for images demonstrating nuclear/cytosolic occupancy of RelA and YAP (Fig. 2D, 2D, 4A, FA-B).

The manuscript relies heavily on immunohistochemistry to quantify proteins, but information on the antibodies used is missing; the methods should note the exact clone and source of primary antibodies. The specificities of the antibodies used should be verified. e.g., For quantifying Piezo1 channels in decidual fibroblasts is shown in Figure 4N. The source and exact clone of Piezo1-specific antibodies are not mentioned; one should be careful in using ion channel-specific antibodies because of the difficulty in generating these antibodies, and often commercial antibodies for ion channels are not highly specific. The sentence "This result is remarkable, showing that chronic mechanical stimuli from scar can even change the expression of Piezo1, suggesting that its activity is likely much higher, resulting in near complete activation of NF κ B signaling" should be toned down.

Dose-dependent inhibition of MafG fluorescence (Figure 7E), by SCH772984 is insufficient to claim changes at the RNA level. Hence "These data strongly suggest that Piezo1 mediated PKC activation increased MafG at both RNA, and protein levels" should be toned down. Again, single-channel images should be shown, including isotype controls demonstrating the specificity of antibodies for immunohistochemistry.

It is not trivial to accurately quantify and compare calcium signaling in cells cultured in two different conditions (Scar Vs. Physio) because most calcium dyes leak out of the cells within hours and inherent dye loading variations. Based on Supplementary Movie 8, how representative is Figure 4D? As noted by the authors, Flou-4 photobleached, most Bleach Correction Plugins in ImageJ manipulate the intensities of images; therefore, basal calcium and F/F₀ quantification need clarification. The exact ImageJ bleach correction algorithm should be mentioned, and the effect of the algorithm on intensities should be verified. Have authors considered a more robust method of quantifying and comparing calcium signaling, e.g., using a ratiometric genetically encoded calcium indicator such as tdTomato-GCaMP6f (<https://www.addgene.org/140188/>)?

Some limitations of this study should be discussed, e.g., 1) Conclusions are primarily based on in vitro model system; in vivo relevance of these results are to be carefully extrapolated. 2) HTR-8/SVneo cell line contains both epithelial and mesenchymal cells (PMID: 28161053); have the authors verified the heterogeneity of trophoblasts and considered the potential implications on the invasion experiments?

Writing could be improved to streamline discussion focused on the current results and make few grammatical corrections (e.g., "enter and re-cellularised"; "prognosticator"; etc.); use bold font for "Extended Fig 3C".

To the Editors

We thank you for instituting a comprehensive, and constructive review of our manuscript “**Aberrant matrix signals at scar cause Piezo1 driven NF- κ B mediated inflammation promoting accreta-like deep placental invasion**” by Du, W et al. We also thank you for providing an extended window to submit our revisions, as obtaining samples for a rare disease like accreta took more time than we anticipated. We have revised our manuscript with many new data which we hope to address the criticisms raised by the reviewers. The manuscript has indeed become much more comprehensive, and thorough as result of the revision. Please find our responses to the individual concerns raised by the reviewers below:

Reviewer #1 (Remarks to the Author):

In this manuscript, Du and colleagues develop a uterine in vitro scar model to delineate how mechanical signals play a role into transforming endometrial fibroblasts into scar forming fibroblasts. Through mainly in vitro collagen assays and some complementary tissue placenta accrete spectrum (PAS) specimens, the authors investigate the role of mechanical stimuli in PAS. Although the authors have developed an interesting in vitro assay to study uterine fibrotic fibroblasts, the study lacks rigor overall. The authors claim that invasion of dESFs is critical in determining the aggressive trophoblast invasion without any in vivo assessment and limited human analysis.

Response: We thank the reviewer for the comments. We have worked with our hospitals to include IHC data from more patients (**Figure 1, Extended Figure 1, and Extended Figure 3**). Furthermore, to address the concern regarding limited human analysis, we utilized scRNAseq data from maternal-fetal interface published from multiple studies (Vento-Tormo et al. Nature 2018 ¹, Arutyunyan et al. Nature 2023 ²). Although these data are from physiologically normal samples, the large number of cells and their inherent variability allow robust statistical assessment of our hypothesis that aberrant mechanotransduction proximal to scar can increase inflammatory transition of the decidual fibroblasts. NF-KBIA (coding I κ B α) and NF-KBIE (coding I κ B ϵ) are NF- κ B target genes, serving as a negative feedback regulatory mechanism³. Using gene-gene correlation analysis, we verified the co-expression of PIEZO1 with NFKBIA and NFKBIE, NF- κ B target genes serving as a negative feedback regulatory mechanism, CXCL8 vs NFKBIA, CXCL8 vs PIEZO1, and YAP1 vs PIEZO1 (**Extended Figure 5H-K and Extended Figure 6E**). In addition, we have further discussed the limitations of the study in Discussion regarding in vitro models and in vivo assessment (In response to R #3's last criticism).

Figure 1C: It not clear how the collagen alignment is assessed with the Masson Trichome staining. No obvious difference is shown between the scar distal and scar proximal in the figure provided. Picrosirius red staining would be a better stain to analyze this. Considering the alignment and the elastic modulus are key to in the pattern of the gels this should be more carefully shown.

Response: We thank the reviewer for the suggestion. We have now performed picrosirius red staining and quantified the collagen orientation using OrientationJ plugin in Fiji/ImageJ. The results are shown in updated **Figure 1C-D** and **Extended Figure 1A**.

Figure 1J: The invasion of the two groups shown at 0 and 24 hours does not appear to show a difference. Can the authors provide more convincing data that the invasion is different?

Response: We have now updated representative figures in **Figure 1J**. We also included new analyses regarding the morphological nature of the trophoblastic invasion fronts, characterized by the depth of stromal invasion by the invasive forks. We found that Scar significantly increased the depth of invasion by trophoblast into dESFs comparing to Physio (**Extended Figure 1F-G**).

Figure 1L: It is not clear why the authors feel that difference in migration is attributed to cytokine production, without any analysis (...‘These data suggested that ESFs decidualized on Scar produce cytokines that recruit HTR8s towards scar’)?

Response: We performed the EVT invasion assay in 3 different settings to probe specifically about the kind of mechanisms that are likely driving increased EVT invasion. In the 3rd setting (**Figure 1G** schematic), trophoblast spheroids were spaced away from the dESFs monolayer with a layer of collagen gel before the initiation of invasion. We recorded the relative nuclear spatial position of trophoblasts to dESFs monolayer after two days of invasion using structured illumination (Apotome) based imaging. Our results showed that trophoblasts are closer to dESFs on Scar than dESFs on Physio matrices (**Figure 1K-L**). Therefore, we concluded that paracrine signals from dESFs plated on Scar matrix chemotactically recruit EVTs. We have however modified the related sentences statement to better address our experimental setups and conclusion (Page 5).

Figure 1M: The authors state an increase in inflammatory genes but only CXCL2 is shown?

Response: We agree that our conclusion based on that figure is premature, and is supported by subsequent figure (**Figure 2 and Figure 3**). We have therefore edited the statement stating only that CXCL8 is increased apart from genes associated with actin contractility (Page 5).

Figure 1N: The authors state that specific cytokines recruit EVTS preferentially towards the scar but this has not been tested?

Response: Our 3D invasion assay (**Figure 1G**) indicates that scar dESFs recruit EVTs by chemoattraction (**Figure 1K-L**). Although we agree that the implication of inflammatory cytokines is not proven till Figure 2, so we have edited the statement and simply state that “mechanical cues presented by the scar matrix alter dESF state to chemotactically recruit EVTs preferentially towards the scar.” (Page 5) In the following Figure, we explored the possible

pathways and show dESFs on Scar matrix chemotactically recruit EVT's via secreted IL-8 and G-CSF. We thank the reviewer for pointing this out, and agree that the conclusion was premature in Figure 1 itself.

Figure 2A: Patient number and demographics are not clearly documented. How many PAS specimens have been included.

Response: We have now included the patient number and have included **Extended Table 1** to show patient demographics. Two more samples from Accio Biobank Online without demographic information are not listed in the table.

Figure 2G: There is no obvious difference in the RelA expression between scar distal and scar proximal in the data provided in 2G. Can staining for other components of the NF- κ B transcription factor complex be stained for including NF- κ B p50 to show a difference?

Response: In **Figure 2G**, although decidual fibroblasts in scar-proximal and scar-distal regions expressed similar level of cytoplasmic RelA, the scar-proximal decidual fibroblasts showed more nuclear RelA than scar-distal decidual fibroblasts, as indicated by the co-localization of RelA and DAPI signals. Indeed, it is the nuclear RelA that is essential for the activation of NF- κ B pathway. Our results showed that scar-proximal decidual fibroblasts expressed more nuclear RelA, which indicated more activation of NF- κ B pathway.

Based on the Reviewer's suggestion, we also performed p50 staining together with RelA (p65) staining. Our results showed the co-localization of p65/p50 in nucleus in scar-proximal decidual fibroblasts (**Extended Figure 3G**), further indicating the nuclear accumulation of the p65/p50 NF- κ B transcription factor complex. We've included p50 staining data in Page 6. We thank the Reviewer for this suggestion.

Figure 3B: The IL-8 staining looks intranuclear and is not clearly different. Why was IL-8 chosen over other cytokines?

Response: We've updated the **Figure 3B** and showed each individual channel of image together with the merged image (**Extended Figure 4A**). It can be seen that IL-8 is enriched in the ER and Golgi. This is because we had used GolgiStop (see **Methods**) (BD Biosciences, Catalog No: 554724) to amplify the IL-8 signal before immunofluorescent imaging. The reason is that immunostaining of secreted ligands is typically challenging as the amount of ligands within a cell is limited owing to continuous secretion (ELISA captures the extracellular secreted ligand, and therefore is a time integral of production, while cellular immunodetection is essentially a snapshot. Preventing secretion by GolgiStop can capture the time integral before compensatory regulations may kick in).

The reason why IL-8 was chosen is because IL-8 encoding gene, CXCL8 is one the most differentially expressed gene on Scar (**Figure 3A**), and has been previously implicated to promote EVT migration⁴, suggesting that EVTs can be stimulated by the ligand likely because they express the requisite receptor, and the necessary signaling.

Figure 3J: The effect of IL-8 knockdown should assess whether the fibroblasts lose their inflammatory phenotype and not only their migration trajectories. For example, is their fibrotic phenotypes at a molecular level affected?

Response: We thank the reviewer for the suggestion. We have now checked α -SMA and vimentin in dESF^{CXCL8-KD}. We found no statistical difference in α -SMA and vimentin expression in scrambled dESFs and dESF^{CXCL8-KD} (**Extended Figure 4B-E**), suggesting that the fibrotic phenotype of the dESFs was not affected by CXCL8 knockdown (Page 7). To note, we have not checked the migratory properties of the fibroblasts after CXCL8 knockdown, but the capacity of these fibroblasts to induce EVT migration (**Figure 3G-H**).

Figure 4N: It is not clear why Piezo1 was chosen above other mechanical pathways? The link between Yap and piezo 1 was not studied ? It was not clear why the authors evaluated YAP?

Response: Because cells sense extracellular mechanical cues through a class of proteins on the plasma membrane known as mechanosensitive ion channels (MSICs), we first ask if dESFs express any MSICs. Gene expression analysis revealed that PIEZO1, which encodes a key mechanosensitive ion channel, Piezo1, was the only MSIC highly expressed and significantly upregulated on Scar vs Physio dESFs (**Figure 4B**) (Page 7), hence it was chosen for further analysis.

To verify the link between Yap and Piezo1, we treated dESFs with 10 μ M LATS-IN-1 (an inhibitor of Yap phosphorylation, thereby promoting Yap nuclear translocation) overnight. LATS-IN-1 treatment elevated nuclear Yap expression (as expected) as well as Piezo1 expression (**Extended Figure 6C-D**). In addition, using single-cell RNAseq data of maternal-fetal interface from Arutyunyan et al. Nature (2023)², we checked YAP1 expression in decidual fibroblasts in vivo. We found the co-expression of YAP1 and PIEZO1 in a specific group of decidual stromal cells (**Extended Figure 6E**). Our results indicate that Yap may play the role of a transcriptional regulator of Piezo1 in decidual fibroblasts (Page 9). We thank the Reviewer for the suggestion.

Figure 4J: Can the authors explain why there is no difference in RelA expression following GsmTX4 and Yoda, only p-RelA?

Response: We showed that Piezo1 activation in dESFs on Scar could regulate NF- κ B activation by checking RelA phosphorylation, a key regulator in NF- κ B activation by enhancing its

transactivation potential^{5,6}. We used total RelA amount as reference to show it is likely the phosphorylation of RelA that contributes to NF- κ B activation. The immunoblot in **Figure 4J** is consistent with the overall hypothesis. We have explained it within the text (Page 8).

Figure 5: The link between NF κ B signaling to glycolysis is not clear. Why was PI3K pathway chosen? Immunostaining or RNA scope has not performed? The evidence supporting Piezo1 and PKC activation is not convincing. The difference in the expression between AKT1, PRKCA, EKR1/2 is not clear.

Response: In Figure 5, we showed that glycolysis provides energy needed to support high contractile phenotype of dESF on scar. To explore the link between NF- κ B pathway to glycolysis, we have now knocked down RelA expression in dESFs and performed 2-NBDG uptake assay. We found that RelA knockdown reduce the 2-NBDG uptake in dESFs (**Extended Figure 6J-K**), indicating that NF- κ B pathway and glycolysis are corelated.

We did not choose PI3K pathway, but the Protein Kinase C pathway, which showed a strong signal in RNASeq pathway analysis (**Figure 6A**), because PKC is regulated by Ca²⁺ signaling modulated by Piezo1. Using substrate immunoblotting for phospho-PKC in dESFs treated with GsMTx-4, a Piezo1 channel blocker, we showed that GsMTx-4 decreased PKC activation (**Figure 6C**). However, GsMTx-4 also inhibits other calcium channels, such as TRPC6. To better assess the role of Piezo1 in PKC activation, we further performed p-PKC substrate immune blotting in dESFs treated with Yoda1, a selective activator of Piezo1. We found that Yoda1 elevated PKC activity (**Extended Figure 7**). Together, our results indicate that Piezo1 is an upstream activator of PKC pathway.

The differences between AKT1, PRKCA, MAPK actually refer to pathway activation score, and not expression, as determined by RNAseq analysis of the underlying gene expression. This was used to provide a candidate selection for pathways which are likely activated on Scar. As discussed above, and in Figure 6, we show that Scar induced Piezo-1 can activate PKC signaling (mediated by Ca²⁺).

Reviewer #2 (Remarks to the Author):

The theme is important. The concept is new and the experimental design was well structured and reasonable. I have minor suggestions.

Response: We thank the reviewer for the favorable receipt of our manuscript and for considering the concept as new, and the experimental design being well structured. We have carefully perused the comments provided by the Reviewer and have made several changes in the text to reflect the suggestions.

1. PAS (placenta accreta spectrum; creta (not accreta!), increta, percreta) should be briefly defined regarding its sub-classifications. To tell things strictly, there is not terminology of “placenta accreta” or “accreta”. FIGO 2018 effort was mainly to extinguish the ambiguity of the term “placenta accreta”, which had two/dual meanings, wide (now, PAS) and narrow (now, placenta creta). Thus, please state/describe “PAS” once you defined classifications, instead of “placenta accreta” or “accreta”. In summary, the term “placenta accreta” no longer exists and thus avoid to use it.

Response: We have revised the manuscript to reflect the correct language. We are very grateful the to Reviewer for bringing the accurate and non-ambiguous current definition to our attention.

2. 1) “PAS pathophysiology” is a “narrower” area compared with “scar pathophysiology” in general. 2) Nature Communications readers are general-medicine > obstetricians. You might, thus, constructed this paper as 1) to determine/identify some aspects of scar physiology, 2) PAS-like model was used, and 3) actually PAS itself is also important pathology recently attracting much wider attention. This is a paper-writing strategy. However, you actually targeted to PAS-model confined to PAS-model); this is “specific” model for PAS. Then, you extrapolated the data to “scar physiology” general. Readers may make question, why PAS to generalize the scar-invasion relationship: for example, A) in the intestinal scar (example; appendectomy scar), the cancer invasion style/manner becomes so different? B) in the episiotomy site, malignant melanoma shows quite (to the extent that comparable to PAS-trophoblast invasion) “different” invasion? There are abundant “parts/ areas” that might indicate the interaction of “scar and invasion”. Some scar may not affect invasion at all, whereas PAS-site affect invasion markedly. Putting this “specific” site and then discuss “scar” general; is this scientifically sound? Of course, this study is not to clarify A) or B): this is not your present intention. What I wish to say is; the present structure is OK? First PAS physiology, then relate (speculate/suggest/exploit) this data with other general condition; this may be much “natural”, I believe. This is based on my personal experience of 495 paper-writing during 4.5 decades. You studied here “PAS” and not “scar general”. I, of course, understand that you CAN speculate/suggest mechanism on scar and cell invasion based on the PAS-model-data. The effort to “change” the paper structure is not difficult: possibly it requires 4-6 hours: only changing the parts before versus after with citation number change. You will hear if you had better incorporate my advice fro EDITOR.

Response: We thank the reviewer for this thoughtful commentary. We have extensively revised the introduction, as well as the discussion (and other changes), to not to appear to generalize the scar-invasion relationship from PAS, which is a specific example. Instead, we formulated our argument, as suggested by the reviewer to present PAS as an example of scar induced dysregulation in epithelia-stroma homeostasis.

The changes are colored differently in the revised manuscript.

We thank the reviewer for this comment. It has allowed to both broaden the scope of the manuscript, sharpening the focus on PAS pathogenesis, while contextualizing it in a broader

question about how scar may result in pathology characterized by spatial dysregulation stroma-epithelia interface.

Please accept our gratitude for this thoughtful comment.

Reviewer #3 (Remarks to the Author):

Du W et al. developed new models to study the pathophysiological manifestations of deep internal scars, e.g., uterine scars due to cesarean surgery. A combination of decidualized endometrial stromal fibroblasts (dESFs) and mechanically distinct tissue scaffolds (Physio Vs. Scar) are used to study the invasion of extravillous trophoblasts (EVTs). Immortalized EVT cells (HTR8/SVneo) are used for assessing trophoblast invasion. Authors show that scar-associated dESFs promote the invasion of HTR8 cells because of increased NF- κ B signaling and secretion of L-8/G-CSF. This, in turn, is mediated through mechanically activated cation channels, Piezo1. Further evidence demonstrates that glycolysis potentially drives the scar-mediated aberrant responses. PKC is identified as a potential messenger for eliciting scar-induced inflammatory phenotype of dESFs. In addition, MAFG is a suggested factor downstream of Piezo1-mediated activation of EVT recruitment cytokine networks. This comprehensive manuscript addresses an essential question of how internal scars modulate tissue functions. My comments are listed below.

Response: We thank the Reviewer for a careful and constructive criticism, and for finding our manuscript as comprehensive one, addressing an essential question about internal scars modulating tissue functions. We address the specific concerns below.

An extensive body of literature shows that Piezo1 is a "cation" channel, not a calcium-selective ion channel. It is stated that "Piezo1 is a membrane stretch-gated Ca²⁺ channel" (Page 7). If Piezo1 channels in decidualized endometrial stromal fibroblasts (dESFs) selectively conduct calcium, authors should show data on calcium ion selectivity by electrophysiology.

Response: We agree with the reviewer's comments. We have modified the sentence into "Piezo1 is a membrane stretch-gated Ca²⁺ permeable channel." (Page 7)

All representative images in the manuscript are composites of two or more channels. Single channel images should be shown along with merged images, particularly for images demonstrating nuclear/cytosolic occupancy of RelA and YAP (Fig. 2D, 2D, 4A, FA-B).

Response: We have included the corresponding single channel images of **Figures 2A, 2D, 4A, and 5A-B** in Extended **Figures 3A, 3D, 5A, and 6A-B**. The rest of single channel images unmentioned are included in **Extended Figures 3E, 4A, and 8A**. Thanks for the suggestion!

The manuscript relies heavily on immunohistochemistry to quantify proteins, but information on the antibodies used is missing; the methods should note the exact clone and source of primary antibodies. The specificities of the antibodies used should be verified. e.g., For quantifying Piezo1 channels in decidual fibroblasts is shown in Figure 4N. The source and exact clone of Piezo1-specific antibodies are not mentioned; one should be careful in using ion channel-specific antibodies because of the difficulty in generating these antibodies, and often commercial antibodies for ion channels are not highly specific. The sentence "This result is remarkable, showing that chronic mechanical stimuli from scar can even change the expression of Piezo1, suggesting that its activity is likely much higher, resulting in near complete activation of NF- κ B signaling" should be toned down.

Response: We thank the reviewer for bringing this important concern. We have now included the origin of antibodies used in the **Extended Table 2**. For Piezo1 quantification, we used Piezo1 (extracellular domain) polyclonal antibody (Proteintech; Cat No: 15939-1-AP). We verified the specificity of Piezo1 antibody by immunofluorescent staining of dESFs and Piezo1^{KD} dESFs (**Extended Figure 5F-G**). The specificity of this antibody has also been verified in the following papers: PMID: 34985971; 30344046; 28199303; 25119035; 35477758, etc. We also performed TrueBlack lipofuscin autofluorescence quencher (Biotium) treatment on the tissue sections to reduce the background signal. Following the reviewer's suggestion, we have toned down the sentence into "This result is noteworthy as it shows that chronic mechanical stimuli from scar can even change expression of Piezo1. Together with increased expression, and higher activity of Piezo1, aberrant matrix on scar results in inactivation of NF- κ B signaling in decidual fibroblasts" (Page 8)

Dose-dependent inhibition of MafG fluorescence (Figure 7E), by SCH772984 is insufficient to claim changes at the RNA level. Hence" These data strongly suggest that Piezo1 mediated PKC activation increased MafG at both RNA, and protein levels" should be toned down. Again, single-channel images should be shown, including isotype controls demonstrating the specificity of antibodies for immunohistochemistry.

Response: We have removed RNA related expression as the reviewer suggested. Single-channel images have been shown, with isotype controls (**Extended Figure 8A**).

It is not trivial to accurately quantify and compare calcium signaling in cells cultured in two different conditions (Scar Vs. Physio) because most calcium dyes leak out of the cells within hours (We performed the imaging 20 minutes after loading completed; we re-imaged the calcium dynamics using calbyte-590) and inherent dye loading variations (We used ion pore to minimized the inherent dye loading variations due to cellular loading capacity). Based on Supplementary Movie 8, how representative is Figure 4D? As noted by the authors, Flou-4 photobleached, most Bleach Correction Plugins in ImageJ manipulate the intensities of images; therefore, basal calcium and F/F0 quantification need clarification. The exact ImageJ bleach correction algorithm should be mentioned, and the effect of the algorithm on intensities should be

verified. (We've included the algorithm into the manuscript) Have authors considered a more robust method of quantifying and comparing calcium signaling, e.g., using a ratiometric genetically encoded calcium indicator such as tdTomato-GCaMP6f (<https://www.addgene.org/140188/>)

Response: We thank the reviewer for this thoughtful criticism. As pointed out, calcium dye, especially Fluo-4AM, is prone to bleaching. Although, we used built-in bleach correction method (simple ratio) in Fiji/ImageJ to account for the photobleaching, to precisely profile calcium dynamics, as the reviewer suggested, we lentivirally transduced EFSs with the 3rd gen. lentivirus for GCaMP6f, and quantified the calcium signals in dESFs on Scar and Physio. The new results were shown in **Figure 4D-F**, **Extended Figure 5D-E**, and **Supplementary Movie 8**. We have previously used the probe on human differentiated cardiomyocytes (Afzal J et al., Cell Reports, 2022), and have found it to be highly quantitative compared to fluorescent dyes, and we thank the reviewer for this suggestion.

Based on the concerns raised by the Reviewer, we also removed the basal calcium data from **Figure 4D** to account for the inherent dye loading variations (although we do find that Scar dESFs showed a higher level of basal calcium signal compared to Physio dESFs, it is very possible that these differences are due to loading differences ---- we thank the Reviewer for bringing this concern to fore). Because the frequency of calcium dynamics is basal value-independent, and the peak/basal ratio depends on the relative ratio in a single cycle of calcium activity not the absolute value, our conclusions are not affected.

Some limitations of this study should be discussed, e.g., 1) Conclusions are primarily based on in vitro model system; in vivo relevance of these results are to be carefully extrapolated. 2) HTR-8/SVneo cell line contains both epithelial and mesenchymal cells (PMID: 28161053); have the authors verified the heterogeneity of trophoblasts and considered the potential implications on the invasion experiments?

Response: We've revised the discussion based on the reviewer's comments. We discuss both the limitation of the study, as well as its translation to broader question about internal scar induced dysregulation of epithelia-stroma interaction.

Following text is added to the Introduction:

There are many examples of pre-existent scars affecting homeostasis at the stroma-epithelia interface associated with invasive outcomes. These include cancers that emerge from the previous scar tissue, including Marjolin's ulcers, a form of squamous cell carcinoma which emerge in 1-2% sites of burn scars¹³, other neoplasms at the site of scar¹⁴, as well as cancer development at the site of keloids¹⁵. Although there are many case reports of scar associated invasive pathologies, epidemiological studies are few, and the mechanistic understanding of scar induced pathogenesis is severely lacking.

Following text is added to the Discussion:

We also acknowledge the inherent limitations of this study. In vitro models have limitations in representing the in vivo nature of pathology. Although reductive analysis offered by in vitro models can allow deeper mechanistic exploration, future studies should test these hypotheses in direct, in vivo models. Secondly, HTR8 cells contain both epithelial and mesenchymal cells. Although as our thesis is based on decidual transformation by scar, the role of HTR8 is limited to their potential to be recruited by dESF produced cytokines, which we have tested in primary HLA-G+ EVT_s isolated from term placenta.

Writing could be improved to streamline discussion focused on the current results and make few grammatical corrections (e.g., "enter and re-cellularised"; "prognosticator"; etc.); use bold font for "Extended Fig 3C".

Response: We've revised based on the reviewer's comments.

References:

1. Vento-Tormo, R. *et al.* Single-cell reconstruction of the early maternal–fetal interface in humans. *Nature* **563**, 347–353 (2018).
2. Arutyunyan, A. *et al.* Spatial multiomics map of trophoblast development in early pregnancy. *Nature* **616**, 143–151 (2023).
3. Yu, H., Lin, L., Zhang, Z., Zhang, H. & Hu, H. Targeting NF- κ B pathway for the therapy of diseases: mechanism and clinical study. *Sig Transduct Target Ther* **5**, 1–23 (2020).
4. Jovanović, M., Stefanoska, I., Radojčić, L. & Vićovac, L. Interleukin-8 (CXCL8) stimulates trophoblast cell migration and invasion by increasing levels of matrix metalloproteinase (MMP)2 and MMP9 and integrins α 5 and β 1. *Reproduction* **139**, 789–798 (2010).
5. Viatour, P., Merville, M.-P., Bours, V. & Chariot, A. Phosphorylation of NF- κ B and I κ B proteins: implications in cancer and inflammation. *Trends in Biochemical Sciences* **30**, 43–52 (2005).
6. Christian, F., Smith, E. L. & Carmody, R. J. The Regulation of NF- κ B Subunits by Phosphorylation. *Cells* **5**, 12 (2016).

REVIEWER COMMENTS

Reviewer #1 (Remarks to the Author):

We thank the authors for providing extensive revisions to their manuscript. We appreciate that the authors added additional patients, improved imaging, and quantification (e.g. based on extent of invasion). They also reworked the phrasing in their results section that has improved readability.

Although most of our comments have been addressed, some were not. For example, the authors provide analysis from scRNAseq data from maternal-fetal interface from two studies; however, these are physiologically normal samples, and therefore are limited in their ability to draw connections to the pathology investigated in their in vitro model. Further, the authors did not complete the request for in vivo assessments to support their conclusion that invasion of dESFs is critical in determining the aggressive trophoblast invasion.

Reviewer #2 (Remarks to the Author):

The authors faithfully reacted to this reviewer's advice, of which incorporation into the revised manuscript has markedly improved the paper quality. Now, the context has become much clearer. I have nothing further to improve the manuscript.

Reviewer #3 (Remarks to the Author):

The authors have addressed the concerns raised by the reviewers. Additional data and modifications to the text are acceptable.

RESPONSE TO REVIEWERS

Reviewer #1 (Remarks to the Author):

We thank the authors for providing extensive revisions to their manuscript. We appreciate that the authors added additional patients, improved imaging, and quantification (e.g. based on extent of invasion). They also reworked the phrasing in their results section that has improved readability.

Although most of our comments have been addressed, some were not. For example, the authors provide analysis from scRNAseq data from maternal-fetal interface from two studies; however, these are physiologically normal samples, and therefore are limited in their ability to draw connections to the pathology investigated in their in vitro model. Further, the authors did not complete the request for in vivo assessments to support their conclusion that invasion of dESFs is critical in determining the aggressive trophoblast invasion.

Response: We thank the Reviewer for the acknowledgement for our revisions. Towards the suggestions further made by the Reviewer, we have now included new data: (i) analysis of scRNAseq data from PAS patients, which was just available this month, strongly supporting our hypothesis that the decidual cells proximal to accreta manifestation show inflammation, as well as (ii) in vivo mouse model showing HTR8 invasion in a matrigel plug embedded with ESFs, and decidualized ESFs.

Specifically, we show confirmation of our hypothesis at multiple levels using reanalysis of publicly available scRNA seq data from accreta, showing that expression of PIEZO1 (**Extended Figure 5B**) and MAFG (**Figure 7D**) are significantly higher in the adherent decidua than in the non-adherent decidua of PAS patients. Moreover, we also found that expression of contractility related genes, including YAP1, MYLK, and MYH9 are all significantly higher in adherent decidua (**Extended Figure 7C and E**). Furthermore, our GSEA analysis showed that NF- κ B pathway is significantly enriched in adherent decidua of PAS patients compared to non-adherent decidua of PAS patients and decidua of normal pregnancy (**Extended Figure 3I-J**). Overall, these analyses confirm the physio-pathological relevance of our in vitro PAS model showing that Piezo1 driven NF- κ B mediated inflammation and PAS are causally linked.

The Reviewer also asked us to confirm that invasion of dESFs is critical in determining aggressive trophoblast invasion in vivo. We therefore have used a specific mouse model to study early stromal invasion, using a Matrigel plug embedded with stromal fibroblasts (dESFs, wild-type or gene perturbed) as well as HTR8 spheroids. As EVT's are not present in rodents, we designed our study to specifically answer the question asked by the Reviewer, and have gone beyond to confirm role of PIEZO1 and MAFG expression in dESFs to control EVT invasion in vivo. Specifically, we subcutaneously injecting high concentration Matrigel containing human EVT spheroids and gene edited dESFs into immune-deficient mice, instead of pursuing a mouse model of PAS. To visualize EVT invasion, we pre-labeled cells in spheroids with CM-DiI, a fixable

membrane dye widely used *in vivo* for cell labeling and tracking. We used high concentration Matrigel due to its high viscosity, which prevents the settle down of heavy spheroids after injection, and its capability of forming a plug right after injection, allowing the following excision. After *in vitro* testing that EVT spheroid integrity is not disturbed by forces during mixing and injection (Extended Figure 6A), we optimized conditions to prevent dissolving Matrigel during PFA fixation (pre-fixing Matrigel plug in Glutaraldehyde to crosslink Matrigel). Afterwards, the Matrigel plug were excised out without any damage (Extended Figure 6B-E). The experimental details are listed in the revised Methods section. We showed using *in vivo* EVT-dESF invasion experiment that knockdown of PIEZO1 and MAFG in dESFs impeded EVT invasion (**Figure 4E and Figure 7K**), consistent with our *in vitro* invasion results. Together with scRNAseq analysis of patient derived data, histological analyses, and *in vivo* experimentation, our data confirm the *in vivo* physiological confirmation of our hypothesis.

Reviewer #2 (Remarks to the Author):

The authors faithfully reacted to this reviewer's advice, of which incorporation into the revised manuscript has markedly improved the paper quality. Now, the context has become much clearer. I have nothing further to improve the manuscript.

Response: We appreciate the reviewer's time and thank the reviewer for the constructive comments.

Reviewer #3 (Remarks to the Author):

The authors have addressed the concerns raised by the reviewers. Additional data and modifications to the text are acceptable.

Response: We thank the reviewer's time and efforts in helping us improving the manuscript.

REVIEWERS' COMMENTS

Reviewer #1 (Remarks to the Author):

In this revision, the authors have responded to our concerns and we feel that the added data have strengthened the manuscript, which we have no remaining issues with.